# Decoupling of a Douglas fir canopy: a look into the subcanopy with continuous vertical temperature profiles

Bart Schilperoort[1], Miriam Coenders-Gerrits[1], César Jiménez Rodríguez[1,4], Christiaan van der Tol[3], Bas van de Wiel[2], and Hubert Savenije[1]

[1]Delft University of Technology, Water Management department, Stevinweg 1, 2628 CN Delft, the Netherlands
[2]Delft University of Technology, Geoscience & Remote Sensing department, Stevinweg 1, 2628 CN Delft, the Netherlands
[3]University of Twente, Faculty of Geo-Information Science and Earth Observation (ITC), Hengelosestraat 99, 7514 AE, Enschede, The Netherlands
[4]Tecnológico de Costa Rica, Escuela de Ingeniería Forestal. 159-7050, Cartago, Costa Rica

**Correspondence:** Bart Schilperoort (b.schilperoort@tudelft.nl)

**Abstract.**

Complex ecosystems such as forests make accurately measuring atmospheric energy and matter fluxes difficult. One of the issues that can arise is that parts of the canopy and overlying atmosphere can be turbulently decoupled from each other, meaning that the vertical exchange of energy and matter is reduced or hampered. This complicates flux measurements performed above the canopy. Wind above the canopy will induce vertical exchange. However, stable thermal stratification, when lower parts of the canopy are colder, will hamper vertical exchange. To study the effect of thermal stratification on decoupling, we analyze high resolution (0.3 m) vertical temperature profiles measured in a Douglas fir stand in the Netherlands using Distributed Temperature Sensing (DTS).

The forest has an open understory (0 – 20 m) and a dense overstory (20 – 34 m). The understory was often colder than the atmosphere above (80% of the time during the night, >99% during the day). Based on the aerodynamic Richardson number the canopy was regularly decoupled from the atmosphere (50% of the time at night). In particular, decoupling could occur when both $u_* < 0.4$ m s$^{-1}$ and the canopy was able cool down through radiative cooling. With these conditions the understory could become strongly stably stratified at night. At higher values of the friction velocity the canopy was always well mixed. While the understory was nearly always stably stratified, convection just above the forest floor was common. However, this convection was limited in its vertical extent; not rising higher than 5 m at night and 15 m during the day. This points towards the understory layer acting as a kind of mechanically 'blocking layer' between the forest floor and overstory.

With the DTS temperature profiles we were able to study decoupling and stratification of the canopy in more detail, and study processes which otherwise might be missed. This type of measurements can aid in describing the canopy-atmosphere interaction at forest sites, and help detect and understand the general drivers of decoupling in forests.

**Keywords.**

canopy; stratification; decoupling; atmospheric stability; distributed temperature sensing; temperature profile;

# 1 Introduction

Measuring atmospheric fluxes over complex ecosystems such as forests has always been problematic due to the height of the roughness elements, which typically extends several tens of meters (Wilson, 2002; Barr et al., 1994). The large roughness layer above a tall canopy also makes it difficult to apply many theories of wall flows as well as to apply and validate traditional similarity theory (Katul et al., 1995). As compared to, for example, a thin grass layer, the tall geometry and internal structure of the forest may allow large turbulent structures within the canopy layer, which will interact with the overlying atmospheric flow (Raupach, 1979). This turbulence may either be generated by wind shear from interaction with the canopy geometry, or be generated and suppressed by local buoyancy effects (Baldocchi and Meyers, 1988). When the air near the surface is warmer than ambient air (and thus less dense), convection is generated. Likewise, when the air near the surface is colder, mixing is suppressed due to the density stratification. These local turbulent exchange regimes will greatly influence the exchange rates of energy and matter away from the forest to the higher atmosphere.

Considering energy and gas exchange from the surface to the atmosphere, the different exchange regimes will cause parts of the canopy to be 'coupled' or 'decoupled' from each other and the atmosphere above. When a canopy is coupled to the atmosphere, exchange of heat and gasses such as water vapor and $CO_2$ takes place between the canopy air and the atmosphere. When a canopy is decoupled from the atmosphere, little turbulent exchange takes place. Different exchange regimes can occur, ranging from a fully decoupled canopy, a partly decoupled canopy, to a fully coupled system where there is turbulent exchange between the subcanopy and the atmosphere (Göckede et al., 2007). These regimes vary per site and are dependent on both the forest structure and the ambient weather conditions.

In particular, nighttime decoupling of the subcanopy is an issue in flux measurements; the so-called 'nighttime (flux) problem' (Aubinet et al., 2012). This usually occurs when the atmosphere is stably stratified and wind speeds are low (Thomas et al., 2017). If the flow above the canopy is (partly) decoupled from the within canopy flow, above-canopy observations are a poor representative of the overall dynamics (Jocher et al., 2017). This will affect the interpretation of on-site flux measurements such as heat, water vapor and $CO_2$ (Fitzjarrald and Moore, 1990). Especially when determining the net ecosystem exchange of $CO_2$, decoupling has to be taken into account (Jocher et al., 2017). In cases where the forest floor is sloped, the combination of decoupling and density flows and the subsequent advective transport can play a big role in the transport of heat and gasses (Alekseychik et al., 2013).

In previous studies decoupling has been determined in a number of ways. A commonly used method has been so-called '$u_*$ filtering' (Goulden et al., 1996; Papale et al., 2006; Barr et al., 2013; Alekseychik et al., 2013), where data with low friction velocities ($u_*$) is flagged. The threshold for $u_*$ is generally based on the sensitivity of the $CO_2$ flux to $u_*$, and can vary in

time. Barr et al. (2013) derived a $u_*$ threshold for varying sites and found a stable threshold value for 28 out of 38 tested sites, albeit with a higher value than a dynamic threshold would have. Ten sites lacked a well defined threshold. Besides the method not being applicable to every study site, $u_*$ filtering also does not take into account any buoyancy forcing (Jocher et al., 2020). To incorporate this, Bosveld et al. (1999) proposed to use an aerodynamic Richardson number, based on based on the friction velocity above the canopy, and the temperature difference between forest interior and atmosphere above. However, determining the decoupling threshold requires a highly accurate air temperature profile above the canopy and radiometric surface temperatures of the canopy, which are generally not available.

To address the shortcomings of $u_*$ filtering, methods have been developed that make use of vertical wind speed ($w$) measured within the canopy. Thomas et al. (2013) introduced a method based on the standard deviation of the vertical wind speed ($\sigma_w$) measured both above and in the canopy. When the canopy is fully coupled, the relationship between above and in canopy $\sigma_w$ is linear. This linear relationship breaks down during decoupling. Jocher et al. (2020) applied telegraphic approximation (TA), where the proportion of the data where the direction of $w$ above and below the canopy have the same direction is used (Cava and Katul, 2009). A high value of TA means that the two air masses are well coupled, while low values indicate decoupling. A second method used by Jocher et al. (2020) is the cross-correlation maximum between above and below canopy $w$, calculated for each flux averaging interval (Foken, 2017).

With measurements both above the canopy and in the subcanopy, a better estimation of the fluxes is possible (Thomas et al., 2013; Jocher et al., 2018). Usually, however, eddy covariance measurements are only available above the canopy. Hence better knowledge on whether the subcanopy is decoupled or not will increase the accuracy of the interpretation of flux data, and consequently forest behavior.

In the past some high density vertically-distributed measurements have been performed in canopies, namely in a walnut orchard (Patton et al., 2011) and in a very open boreal forest (Launiainen et al., 2007). Several sonic anemometers were distributed along the height of the canopy. However, in both cases the canopies were very open, and decoupling was not an issue at these sites. The main focus of the studies was boundary layer parameterization and profiles of turbulent statistics. Instead of considering discrete point observations along the height of the canopy, we search for a more continuous probing of temperature to get a more detailed view on the influence of static stability on decoupling along the entire height of the canopy.

By using distributed temperature sensing (DTS) technology (Smolen and van der Spek, 2003; Selker et al., 2006), it is possible to measure temperature with a high spatial resolution (30 cm) using a single fiber optic cable. If this cable is placed vertically along a flux tower, a full temperature profile from the forest floor to above the canopy can be measured. As the entire cable is calibrated continuously, it can be used to accurately measure small gradients (Schilperoort et al., 2018; des Tombe

et al., 2018; Izett et al., 2019). Additionally, the cable can be installed in a coil configuration to measure at even higher (<1 cm) spatial resolutions (Hilgersom et al., 2016).

With these high resolution temperature profiles we can study the response of the atmosphere-canopy system, and study if vertical mixing by turbulence is suppressed or enhanced due to thermal stratification.

## 2 Materials and Methods

### 2.1 Study Site

The measurements were carried out at the 'Speulderbos' research site in Garderen, The Netherlands (52°15'N, 5°41'E, Fig. 1). A 48 m tall measurement tower is located within a patch of Douglas fir trees (*Pseudotsuga menziesii (Mirb.) Franco*), surrounded by a mixed forest consisting of patches of coniferous and broadleaved trees. Douglas fir trees were planted on the site in 1962, and they have since grown to be ~34 m tall (Cisneros Vaca et al., 2018a). Actual tree density is 571 trees per
90 hectare, with a mean trunk diameter at breast height of 35 cm (Cisneros Vaca et al., 2018b). The leaf area index, measured with a LI-COR LAI 2000 Plant Canopy Analyzer, is approximately 4.5 $m^2m^{-2}$ (Cisneros Vaca et al., 2018b). At the site only some sparse undergrowth is present (mosses, ferns), and most of the forest floor is covered by litter (Fig. 2a). A profile of the plant area index is shown in Appendix A. The canopy structure, uniformity of tree heights, and lack of sparse undergrowth is typical for Douglas fir plantations across Western Europe, Canada and the Western United States (Schmid et al., 2014; Winter
et al., 2015; Douglas et al., 2013). The surrounding forest varies in age and height, and is intersected by access roads, which creates gaps in the canopy. As such, the overall area is heterogeneous on a scale of one kilometer (Fig. 1). The area is slightly undulating, with a local grade of ~2.5%.

The canopy at the Speulderbos site is tall with a distinct vertical structure. To further study the temperature gradients in the canopy, we split the profile in multiple sections as follows (based on Parker (1995) and Nadkarni et al. (2004), illustrated in
Fig. 3:

- Above-canopy: includes the air mass located above the canopy layer, up to 48 m where the vertical fluxes are determined using the eddy covariance measurements.

- Overstory: consists of virtually all the branches with photosynthetically active needles, located between 34 m and 20 m. This layer is fully illuminated from above and all branches receive direct sunlight. From 20 m downwards live branches
are almost absent, and the present branches are dead remnants of earlier growth stages. For further analysis we define three sections of the overstory:

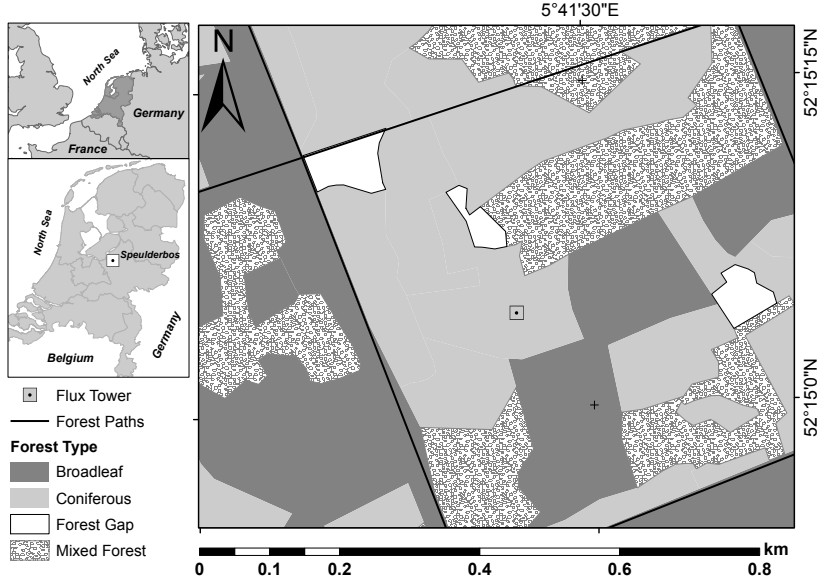

**Figure 1.** Forest type distribution around the tower site at the Speulderbos forest, the Netherlands.

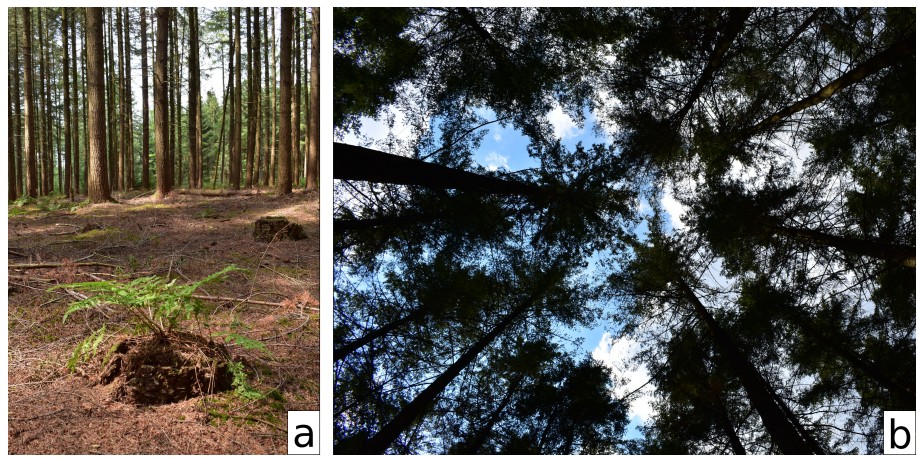

**Figure 2.** The forest floor, showing the open understory (a) and a sky view from below (b), showing the more dense canopy top. Photos taken in July 2018.

– Tree tops: top of the overstory, from 30 to 34 m, occupied by the tops of the tallest trees, but otherwise absent of vegetation.

– Central overstory: between 25 and 30 m, starting at the most dense part of the canopy where most of the solar radiation is absorbed, to 30 m. Nearly all branches are fully illuminated.

- Lower overstory: from 20 to 25 m, dominated by dense branches which are partially shaded by the leaves/needles above.

- Subcanopy: the section between the ground and overstory, and consists of three sections:

- Upper-understory: is composed by a vertical section dominated by dead branches between 10 m to 20 m height.

- Lower-understory: comprises the region between 1 m to 10 m of the forest stand. It includes the section dominated by bare tree stems, without branches or bushes.

- Forest floor: the lower section along the forest canopy from 0 m to 1 m. At the Speulderbos site, it is dominated by the presence of organic debris (litter), mosses attached to the debris, and ferns scattered around the plot.

While there is nearly no horizontal wind at all in the center of the overstory, the open understory at the site has a mean wind speed of 0.8 m s$^{-1}$, and wind speeds of up to 1.5 m s$^{-1}$ are common. The ground heat flux at 1 cm depth ranged typically between -15 and 15 W m$^{-2}$, and is dominated by variations in daily mean temperature rather than by the diurnal cycle. Below the canopy, net radiation was typically between -5 and 25 W m$^{-2}$, predominantly positive/downward. The sensible heat flux at 1 m was mainly between -10 and 5 W m$^{-2}$, and shows a distribution which is highly skewed towards negative (i.e. downward) heat fluxes.

## 2.2 Setup

The temperature of fiber optic (FO) cables was measured for 250 days between 2015 and 2018 using the DTS technique (Selker et al., 2006). DTS measurements are made by shooting a laser pulse down a FO cable, and analyzing the backscattered light. Some of the backscattered light will have undergone Raman scattering. As Raman scattering is sensitive to temperature, it can be used to determine the temperature of the fiber (Smolen and van der Spek, 2003). The location of the temperature measurements along the fiber is determined using the time-of-flight of the laser pulse.

From the DTS machine a fiber optic cable was routed through a calibration bath, up to the top of a 46 m tall scaffold tower, down along the tower, and back to the calibration bath (Fig. 3).The cable was guided by PVC rings secured to horizontal wooden beams. The part of the cable closest to the tower was a wetted cable, for the determination of the wet bulb temperature (not used in this study). The cable further from the tower was used for the air temperature measurements, and was kept at a distance of ~1.2 m away from the tower. To increase the measurement resolution closer to the forest floor a coiled cable was used (Hilgersom et al., 2016). A cable was routed through the calibration bath, over the forest floor to a coil configuration, and then back to the calibration bath. The coil contains 8 m of cable in a coil of 1 m height.

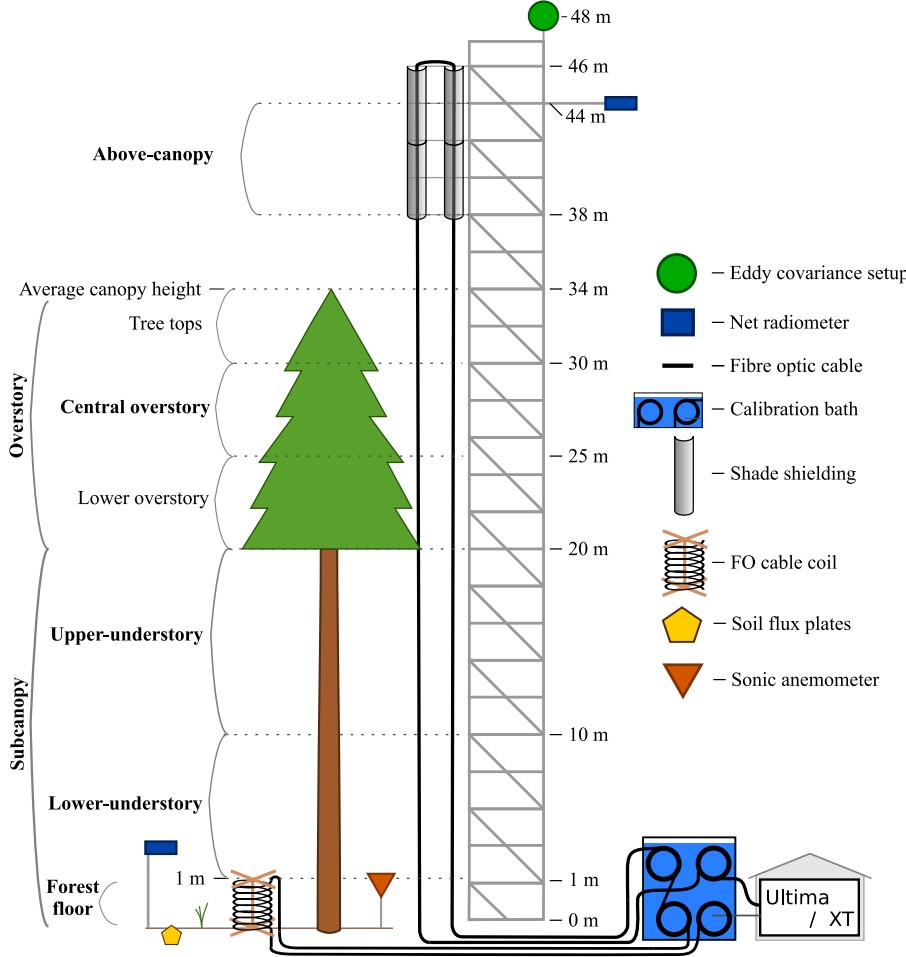

**Figure 3.** Schematic overview of the measurement setup at the tower. On the left the different sections are shown. There is a gap in the sections from 34 to 38 m, as the unshielded DTS data there is not reliable.

Both cables are 6 mm in diameter, with braided steel wire and a wrapped stainless steel coil around the core, and coated with PVC. The integration time of the DTS device was set to 1 minute, as the response time of the FO cables was up to 5 minutes in air. While the cables have a slow response time, they are robust and able to survive for years without needing replacement. This makes long-term measurement easier than with a less protected (thinner) fast response cable. The FO cables were connected to a Silixa Ultima-S DTS device (Silixa Ltd., Elstree, UK), or to a Silixa XT-DTS from February to March 2018. The FO cables were spliced together and placed in a single ended configuration. To calibrate the cables a calibration bath at ambient temperature with a Pt100 resistance thermometer was used. The water bath was kept well mixed using an aquarium pump.

Calibration was performed using the Python package *'dtscalibration'* (des Tombe and Schilperoort, 2019; des Tombe et al., 2020).

When measuring air temperature with DTS, direct sunlight will warm up the FO cable, which will therefore deviate from the air temperature. To shield the above canopy section of FO cable from direct sunlight, shade shielding was placed from 38 m to 46 m (Schilperoort et al., 2018). The shielding was not present in the years 2015 and 2018, however a comparison to reference sensors showed that while the daytime gradients were overestimated compared to reference sensors, the absence of shielding did not have a large effect on the measured stability. Below the canopy, direct sunlight rarely reaches the cables.

Beside solar radiation, FO cables are affected by radiative cooling. To estimate the magnitude of the error, a comparison between a similar setup and reference sensors has been made (Appendix B). While the environments are different, we assume a similar error estimate because the meteorological conditions (that determine the radiative cooling) are comparable. Under the conditions encountered in the understory, the absolute error will be between 0 and 0.10 K. Errors in the gradient will be up to 0.01 K m$^{-1}$. At the top of the canopy the error will be larger, as the cable is exposed to the sky. The gradients were not corrected or adjusted, but left as is.

As the full vertical profile is measured right next to the tower, the tower itself will have an unknown influence on the measurements. The tower itself has a scaffold structure, with a base size of 3.7 by 2.1 m. The scaffold structure is made from 47 mm diameter stainless steel tubes.

At the top of the tower (48 m) an eddy covariance (EC) system was installed, consisting of a Campbell CSAT3 sonic anemometer and a LI-COR Biosciences LI7500 gas analyzer, logged at 20 Hz. The raw data from the eddy covariance system was analyzed using LI-COR's EddyPro® software (LI-COR Inc., 2016). The combined quality flag system from Foken et al. (2004) is used. Only the fluxes with a quality flag of 0 or 1 are used in this research. These flags are the best quality fluxes and represent fluxes suitable for general analysis, based on steady state tests, integral turbulence characteristics and horizontal orientation of the sonic anemometer. The eddy covariance system was installed on the South West corner of the tower. To account for influences of the tower, all EC data coinciding with a wind direction between 350 and 80 degrees was removed.

At 0.8 m a Gill Instruments Windmaster Pro 1352 sonic anemometer measured 3-dimensional wind speed and sonic temperature, however due to equipment malfunctioning it was only available for a 20 day period in December 2017. This period did not overlap with the DTS or EC equipment. A Kipp & Zonen CNR4 at 44 m, and a CNR1 at 2 m measured the incoming and outgoing short- and longwave radiation. Heat flux plates (Hukseflux HFP01) were installed at a depth of 1 cm ($G_{1cm}$) and a depth of 8 cm ($G_{8cm}$). Weather stations were installed in the understory and in a forest gap 300 m from the tower. Temperatures were measured inside two trees between 19 August 2017 and 6 October 2017. the temperature probes were installed at a depth

of 1 and 2 cm below the bark. Using the temperature difference we determined the biomass heat flux. The biomass heat flux in the bottom 20 m of the forest was in the range of -2.5 to 2.5 W m$^{-2}$. Besides characterization of the field site these sensors are not used in further analysis.

Not all sensors were available for the full measurement period. The DTS measurements were done intermittently, during late summer / early fall in 2015, 2016 and 2017, from January until April 2018 and in June 2018.

## 2.3 Method

To characterize the effect of thermal stratification on turbulent mixing regimes, we calculate the *local static stability* of the potential temperature profile. Static stability of the atmosphere is related to the local temperature gradient. When the temperature gradient is negative, i.e. $\frac{\partial \theta}{\partial z} < 0$, the air closer to the surface is warmer, and natural convection will transport heat upwards. As such it is unstably stratified. When the temperature gradient is positive, the air closer to the surface is colder, no natural convection takes place and turbulent mixing by wind is suppressed. This makes the air stably stratified. When there is no temperature gradient the stability is neutral.

To characterize the *dynamic stability* of the atmosphere, both the effect of thermal stratification and mixing by wind shear have to be taken into account. The ratio of the buoyancy and shear forces can be described using the Richardson number. Following the definition of Bosveld et al. (1999), the aerodynamic Richardson number for decoupling can be calculating using the temperature difference between the air above the forest and the understory:

$$\mathrm{Ri}_A = \frac{gh}{T} \frac{\theta_h - \theta_i}{u_*^2} \tag{1}$$

where $g$ is the gravitational acceleration (9.81 m s$^{-2}$), $h$ is the height at the top of the canopy (m), $T$ is the absolute temperature in the subcanopy (K), $\theta_h$ is the temperature at the top of the canopy (K), $\theta_i$ is the temperature in the subcanopy (K), and $u_*$ is the friction velocity (m $^{-1}$). When $\mathrm{Ri}_A > 1$ buoyancy dominates the flow, when $\mathrm{Ri}_A < 1$ shear dominates the flow. To determine when the canopy would become decoupled, Bosveld et al. (1999) calculated the difference between $\theta_h$ and the radiative surface temperature of the canopy, as well as the aerodynamic surface temperature. The aerodynamic surface temperature is derived from extrapolating the roughness-sublayer temperature profile to the height of the canopy. By comparing $\mathrm{Ri}_A$ to the difference between the temperature above the canopy and both the radiative surface temperature and the aerodynamic surface temperature, an inflection point was found at $\mathrm{Ri}_A \gtrsim 2$, where the aerodynamic and radiative temperatures diverged and decoupled was assumed to occur.

The friction velocity $u_*$ can be calculated as follows (Stull, 1988):

$$u_* = (\overline{u'w'}^2 + \overline{v'w'}^2)^{\frac{1}{4}} \tag{2}$$

where $\overline{u'w'}$ and $\overline{v'w'}$ are the covariance between both horizontal wind speed components and the vertical wind speed component (m s$^{-1}$).

Also, we utilize the so-called parcel method (Thorpe et al., 1989) in order to estimate the vertical extent that an air package will rise due to natural convection in a steady-state environment. The height to which, e.g., a parcel of air from the forest floor will rise is the height at which the local (potential) temperature $\theta(z)$ exceeds the temperature of the forest floor parcel.

In all analyses we only make use of the dry adiabatic lapse rate. Condensation of moisture can contribute to convection in forests (Jiménez-Rodríguez et al., 2020), but this effect has not been taken account in this study due to lack of accurate data.

## 2.4  Data Processing

To accurately determine the stability, we need to make use of the entire profile over which we estimate the stability (Schilperoort et al., 2018). To achieve this we fit the data points of each section to a second order polynomial, minimizing the squared error. The temperature gradient is calculated from the profile fit, and consequently, the potential temperature lapse rate is computed (Kaimal and Finnigan, 1994):

$$\frac{\partial \theta}{\partial z} \approx \frac{\partial T_a}{\partial z} + \Gamma \approx \frac{\Delta T_{a,\text{fit}}}{\Delta z} + \Gamma \tag{3}$$

where $T_{a,\text{fit}}$ is the fit temperature, and $\Gamma$ is the dry adiabatic lapse rate (~0.0098 K m$^{-1}$). The polynomial fits were calculated separately for each section (e.g., lower-understory). The average root-mean-square error of the polynomial fits was ~0.1 K main profile, and 0.02 K for the forest floor coil profiles.

To split up the data into the three conditions; stable, (near-)neutral and unstable, we defined the neutral class for a finite interval of gradients between -0.01 and 0.01 K m$^{-1}$ (the same order of magnitude as the lapse rate).

For the calculation of the aerodynamic Richardson number, we used the 10 m DTS temperature as the canopy internal temperature and the 44 m temperature as the top-of-canopy temperature. The 10 m temperature is in the center of the understory, and therefore represents the general temperature of the understory well. It could be possible to use a profile integrated temper-

ature but this will not change the results significantly. Note that the cable at 44 m height is shielded against direct sunlight, in contrast to the cable between 34 and 38 m. The friction velocity measured at 48 m was used.

To split data between daytime and night we have to define these relative to the local time of sunrise and sunset. To account for the uncertainty around dusk and dawn, we define daytime as starting one hour after sunrise and ending one hour before sunset. Nighttime starts one hour after sunset and ends one hour before sunrise. Local sunrise and sunset times for the measurement site were determined using the ***Pysolar*** Python package (Stafford, 2018). We removed all data points both during rainfall and 60 minutes after rainfall (32% of DTS data), to allow for the understory sensors to be fully dry.

## 3   Results

### 3.1   Characteristic temperature profiles

As a demonstration, two typical profiles are shown in Fig. 4. 15 October 2017 was a sunny day, causing the overstory to heat up due to solar radiation. In the subcanopy the air stayed cooler, and the profile within the canopy is stable to near-neutrally stratified, with the coldest point at the forest floor. During the night there was strong radiative cooling at both the central overstory and the forest floor due to radiative cooling. Note that also the forest floor is able to cool through longwave radiation as it partly 'sees' the open sky (Fig. 2). This will cause a stable stratification above the canopy and above the forest floor, while the bulk of the canopy (2 - 26 m) is unstably stratified due to the colder air in the overstory. On 11 October 2017, an overcast and humid day, the canopy was only slightly warmer during the day, and the entire profile was near neutral during the night. Animations of the temperature profiles are available on Zenodo (Schilperoort et al., 2019).

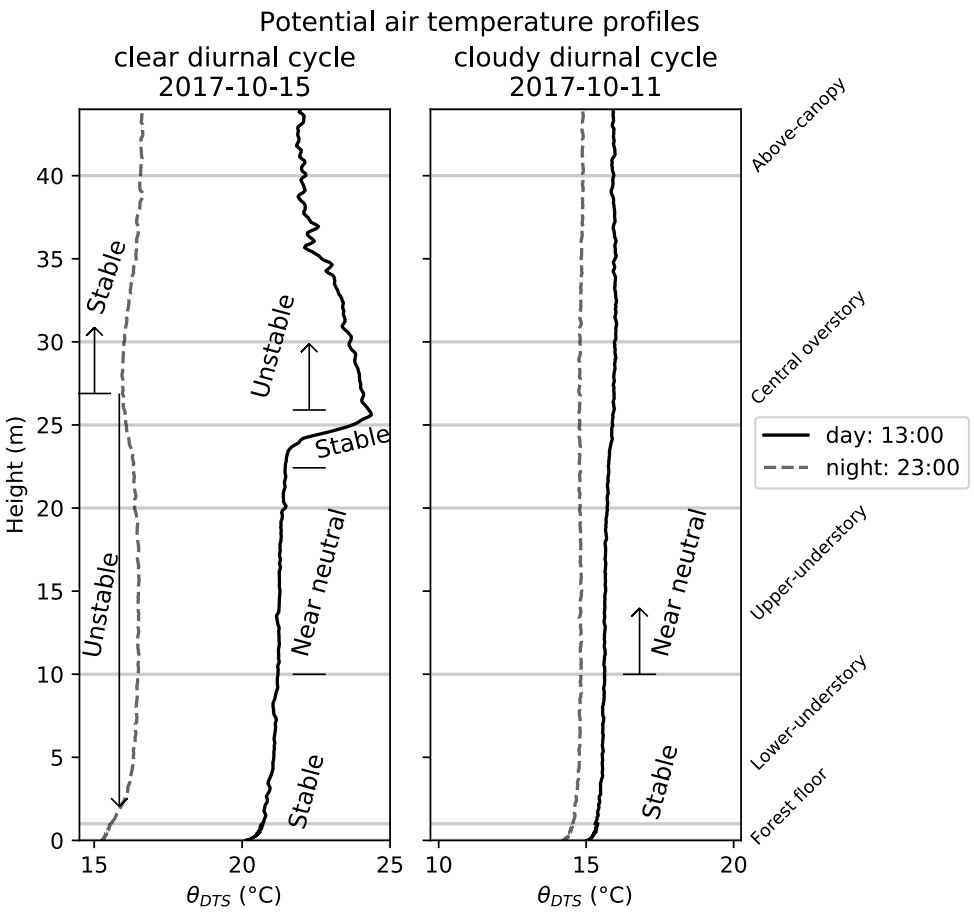

**Figure 4.** Example of potential air temperature ( $\theta(z)$ ) profiles to illustrate the DTS measurements (15 minute average temperature). 15 October 2017 (left) was a sunny day and a clear night. 11 October 2017 (right) was a cloudy day and night. Times are in UTC+1

## 3.2 General climatology: temperature gradient statistics

In order to generalize the features observed in Fig. 4, an in-depth statistical analysis of the local gradients in terms of external forcings was made for the full data set. To this end we grouped the DTS gradients in bins of day and night. The bins are split over the four seasons. For each of these bins, the occurrence of each local stability condition is summed up and compared to the total amount of data points in the bin. This shows the relative occurrence of stable/neutral/unstable conditions (Fig. 5).

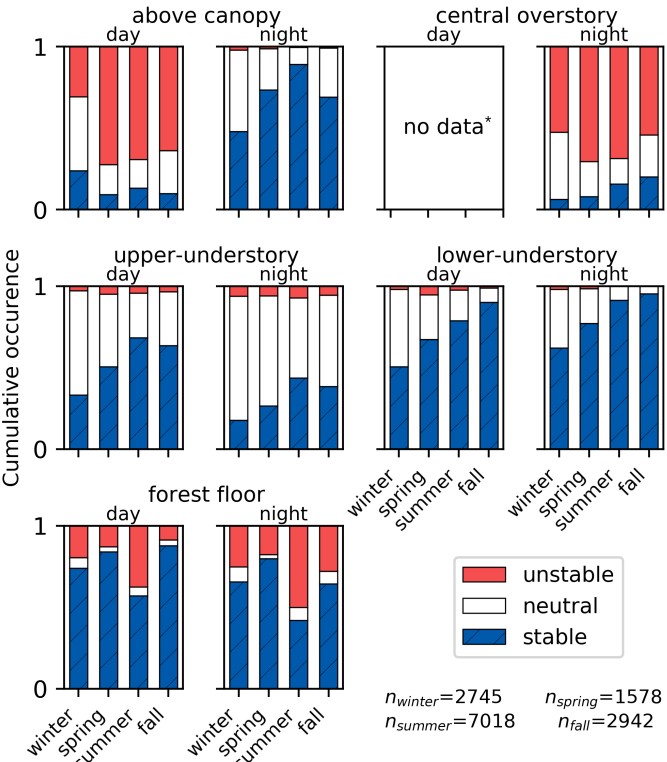

**Figure 5.** Comparison of the local static stability of each cable section (above-canopy, central overstory, upper-understory, lower-understory, forest floor) over time. Y-axis shows the cumulative occurrence of stable/neutral/unstable conditions. Results are split over daytime (left) and nighttime (right), and aggregated over the different seasons. *Central overstory daytime data is not shown due to errors created by solar radiation.

The gradient above the canopy shows the expected diurnal pattern, being mostly unstable during daytime and stable at night. In contrast, the other sections do *not* show this strong diurnal pattern. This discrepancy is an indication that the system in the canopy is often 'decoupled' from the flow above (at least partly), due to the geometry of the vegetation itself, with most of the biomass in the top.

The upper-understory section is mostly stable during daytime and neutral at night. The lower-understory section is nearly always stable, both during daytime and at night. The forest floor section can be both stable and unstable, both during daytime and at night. This may result in persistent mechanical 'blocking' of buoyant air parcels (see below). The unstable gradient at the forest floor section means that convection takes place locally, but due to the stable stratification of the understory this convection would have to travel against the stable gradient in the lower-understory to reach the overstory or atmosphere above the forest. These counter-gradient fluxes are still possible and are likely to occur during larger sweeps (Denmead and Bradley, 1985), where large scale motions are responsible for transport.

However, the results suggests that the stable conditions in the overstory act as a capping inversion for the buoyancy in the lower understory. The question is, what is the actual vertical extend of the convection driven by surface heating. To study this in more detail we used the parcel method to calculate the maximum height for a floor parcel to rise by convection. Figure 6 shows that at night convection from the forest floor rarely exceeds 5 m in height. During the day convective air parcels can rise higher. In 5% of the daytime data they reach 15 m in height, possibly due to sunlight warming up the forest floor. Most likely this occurs at high solar angles during summertime (Fig. 5).

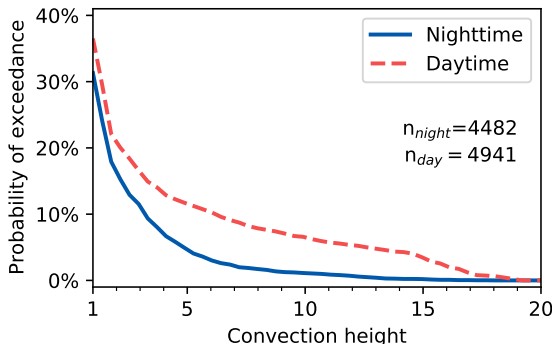

**Figure 6.** Cumulative probability distribution of the convective rising height of a parcel of warm air from forest floor.

From this data it can be expected that there is barely any direct convective transport from the forest floor to the top of the canopy. Also, mixing generated by ambient wind shear over the canopy is largely damped by the thermal stratification of the understory. This may have implications not only for heat transport but also e.g. for the transport of water vapor, $CO_2$ and trace gases. In the next section the effect of turbulence of the vertical mixing is explored.

### 3.3   Decoupling: the Aerodynamic Richardson number

To characterize decoupling of the understory, both the effects of buoyancy and shear have to be taken into account. To this end we utilize the Richardson number as defined in Eq. (1). We restrict ourselves to nighttime cases, which are predominantly influenced by local wind shear above the canopy, and hence by $u_*$.

A cumulative distribution function of the aerodynamic Richardson number is shown in Fig. 7. Only the positive Richardson numbers are shown. However, ~22% of the data points have a negative aerodynamic Richardson number, meaning that the forest interior is warmer than the air above the forest and thus not decoupled. According to Bosveld et al. (1999), decoupling occurs when the aerodynamic Richardson number exceeds approximately 2. In our case this implies that decoupling occurs ~50% of the time at night, showing that decoupling is common at this study site at night.

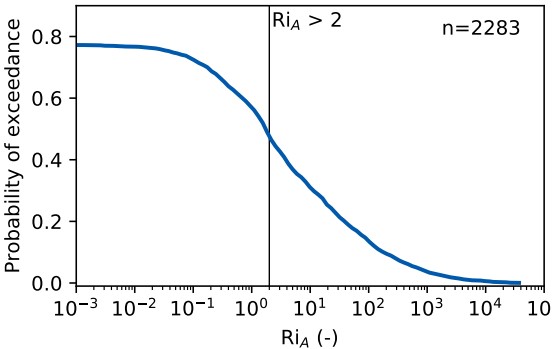

**Figure 7.** Cumulative probability distribution of the aerodynamic Richardson number, for nighttime data. Negative $Ri_A$ values are represented in the maximum of 78% probability of exceedance, i.e., ~22% of the nighttime $Ri_A$ values are negative. Vertical line shows $Ri_A = 2$, which was defined as the 'decoupling threshold' in Bosveld et al. (1999).

The aerodynamic Richardson number was derived by Bosveld et al. (1999) for nighttime conditions, and is not valid under daytime conditions where the friction velocity will be strongly affected by turbulence generated by convection from the top of the canopy. The suppression of mixing by the stable stratification of the understory during daytime is also not taken into account in the aerodynamic Richardson number.

## 3.4    Influence of shear and buoyancy on decoupling

**3.4.1    Magnitude of the temperature difference between the subcanopy and atmosphere**

While wind shear over the canopy will induce mixing, a temperature difference between the atmosphere can either suppress or drive mixing. To study the buoyancy forcing, we compare the temperatures at 44, 10 and 2 m height (Fig. 8). In ~78% of the available data the middle of the understory was colder than the air above, despite the proximity to the biomass and soil. This was previously observed at this site by Bosveld et al. (1999). Closer to the ground, at two meters height, the air was generally

even colder. During the day, the understory was nearly always colder (~99% of the available data). This was to be expected as most incoming sunlight is absorbed at the top of the canopy, heating up the atmosphere above as well, while the understory stays cool.

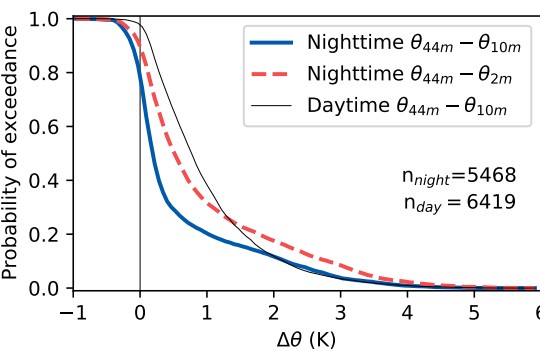

**Figure 8.** Cumulative probability distribution of the difference in temperature between the forest interior and the air above the forest (at 44 m). Solid line shows the nighttime distribution for 10 m, dashed line shows the nighttime difference for 2 m, thin line shows the daytime difference at 10 m. Positive numbers denote a colder forest interior.

### 3.4.2 The relationship between the ambient friction velocity and the local static stability

While stable thermal stratification can suppress mixing, wind shear at the top of the canopy will enhance mixing. Therefore
we compare the above canopy friction velocity with the in-situ observed temperature gradient at several heights (Fig. 9). To relate the $u_*$-temperature gradient relationship to decoupling, the aerodynamic Richardson number is shown by coloring the plot markers.

At night the gradient above the canopy is inversely related to $u_*$, as expected (Fig. 9a). At high shear conditions the air above the canopy is well mixed, resulting in small temperature gradients and a coupled canopy. At low shear conditions it is possible
for the top of the canopy to cool considerably, allowing strong local gradients to occur.

Interesting, the understory gradients (Fig. 9b, c) show a characteristic L-shaped behavior with a kind 'threshold' value for $u_*$: below $u_* \approx 0.4$ large gradients are able to occur, while small gradients are observed for large $u_*$. This threshold value corresponds with the $u_*$ values associated with decoupling of evergreen needleleaf forests found by Barr et al. (2013), although the $u_*$. threshold is strongly site specific, and it is not always possible to derive a $u_*$. threshold. Especially the upper-understory
gradient shows a distinct split between the two coupling regimes; when the canopy is coupled ($Ri_A < 2$), the local gradient is grouped tightly around 0. When the canopy is decoupled, strong gradients can form, with $u_*$ values ranging between 0 and 0.4 m s$^{-1}$.

The relationship between $u_*$ and the forest floor gradient is less clear (Fig. 9d), however the Richardson number highlights the positive correlation: The forest floor is unstably stratified when $u_*$ is low and the canopy is decoupled, and stably stratified
when $u_*$ is high and the canopy is coupled. The relationships between the understory gradients and friction velocity show that the temperature gradients can serve as a proxy for decoupling; when the understory is strongly stably stratified the canopy

is decoupled. However, the understory can still be decoupled even without strong thermal stratification, as shown by the data points in the lower left corners of Fig.9b, c. When the friction velocity is below ~0.2 m s$^{-1}$, the canopy is always decoupled. It is likely that at very low friction velocities the wind will not be able to mix the canopy even though there is no strong temperature gradient (e.g., very low wind, overcast conditions).

While at night turbulent mixing is driven by wind shear (hence friction velocity), during daytime convection generated at the top of the canopy is also important for creating turbulence. Indeed, as shown in Fig. 9e, f, g, and h, the dependence of local temperature gradients on (externally driven) $u_*$ is less well defined and sometime even absent. Unlike the nighttime data, the lower- and upper-understory gradients have no clear threshold or correlation at all.

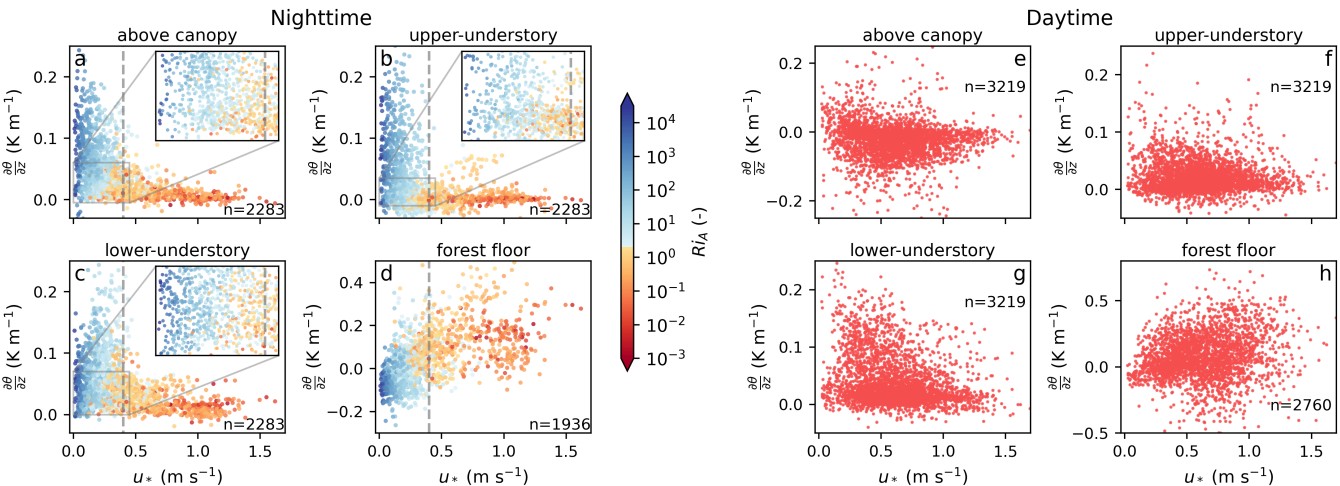

**Figure 9.** Comparison between the friction velocity at 48 m, and the DTS measured temperature gradients. For the nighttime data the markers are colored by the aerodynamic Richardson number, where values of $Ri_A < 2$ (coupled) are colored red, and values of $Ri_A > 2$ (decoupled) are colored blue. The gray vertical dashed line indicates $u_* = 0.4$. Subplot a & e show the above canopy gradient, b & f the upper understory, c & g the lower-understory, and d & h the forest floor gradients.

## 3.5 Discussion

The results show that the subcanopy is nearly always stably stratified, both during the day and at night, and will primarily be decoupled during low-wind conditions. The gradient above the forest floor may become unstably stratified, but convection does not rise high enough to penetrate the overstory. The understory shows consistent decoupling and seems to act like a kind of mechanically 'blocking layer' between the forest floor and overstory. In the overstory, nighttime convection is common. Heat stored in the leaves, branches, and trunks warms up the air in the lower overstory, causing within-canopy convection.

This results in four typical exchange regimes observed at the Speulderbos, schematically illustrated in Fig. 10. Two daytime (a and b), and two night-time (c and d) situations. Fig 10a displays the daytime high wind shear regime. The wind is strong

enough to penetrate into the subcanopy and mix the entire canopy. In Fig 10b wind shear is not strong enough, and the subcanopy is decoupled. Convection can take place above the forest floor but does not progress further upward into the canopy.

At night, exchange between both the subcanopy and top of the canopy, and the atmosphere is dominated by wind shear. In Fig. 10c wind shear is strong enough to mix the entire canopy and prevent strong stable stratification. Local convection can take place within the canopy due to heat released by the leaves, branches and trunks. Fig. 10d the wind is not strong enough to enter the subcanopy, and the subcanopy is stably stratified. Convection from the forest floor is possible, but does not reach the overstory. For nighttime, the subcanopy is decoupled in approximately 50% of the available data. Convection above the forest

floor takes place in ~50% of the low wind shear conditions.

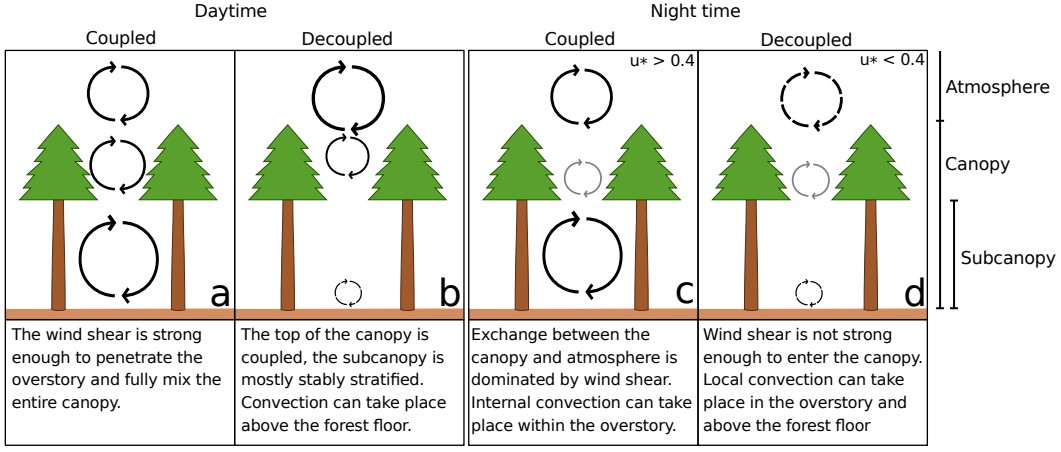

**Figure 10.** Typical exchange regimes observed at the Speulderbos site. The cyclic arrows denote convection/non-local transport.

As explained, wind shear above the canopy has a large influence on the thermal stratification of the canopy. We observe strong stratification at night when $u_*$ is below 0.4 m s$^{-1}$, which is similar to previous research (Barr et al., 2013; Papale et al., 2006). It is important to note that this is not necessarily a fixed threshold, but that some form of hysteresis might be present (van de Wiel et al., 2017). When the subcanopy is already strongly thermally stratified, more wind shear is needed to mix it,

while a canopy which is not stratified will stay mixed more easily. It would be interesting to explore the impact of understory stratification on the friction velocity threshold value, by assessing effects of conditional sampling.

We note that effects of heterogeneity and advection may also play a role in the convective coupling between the forest floor and the overstory. Localized transport (consisting of convective plumes (Jiménez-Rodríguez et al., 2020)) could transport heat from the forest floor through the canopy. As this transport is very local, the measured 15-minute mean vertical profile might

not be representative for the entire forest. Alekseychik et al. (2013) found that drainage flows, where dense cold air flows down slope, can be of influence for decoupling. This could be an important mechanism at this site as well, as the slope of the forest

floor is non-negligible; approximately 2.5%. Due to the open understory, advection in the subcanopy could be an important process, with wind speeds ranging from 0.5 to 2 m s$^{-1}$, and gusts of up to 4 m s$^{-1}$. While it is possible to asses advection and drainage flows if a sonic anemometer is located near the forest floor (Staebler and Fitzjarrald, 2004), we did not study this further due to a lack of sonic anemometer data.

Another limitation of this study is the low measurement frequency (which is limited by the response speed of the fiber optic cables). Any exchange mechanisms that have a timescale under 10 to 15 minutes, such as sweeps or ejections (Gao et al., 1989), will be missed.

## 4    Conclusions and recommendations

In this study we used vertical DTS profiles to study the thermal stratification and the potential of decoupling within the forest canopy. We found that on the Speulderbos measurement site stable stratification of the subcanopy is the dominant state over multiple years and seasons, even though convection can take place just above the forest floor both at night and during the day. The local convection above the forest floor rarely exceeded 5 m height at night, and 15 m during the day, and did not reach the overstory.

Local temperature gradients in the understory were nearly always stable, and showed no strong diurnal pattern. The temperature gradient above the forest floor was stable ~70% of the time, and did not show a strong diurnal pattern either. Besides the stable temperature gradients, dynamic stability indicators such as the aerodynamic Richardson number also indicated decoupling of the understory, up to 50% of the time at night. The air temperature of the subcanopy was mostly colder than the air above the forest, at night (~78%) and especially during the day (~99%). When comparing the temperature gradients to the friction velocity, we found that at night decoupling could occur when $u_* < 0.4$ m s$^{-1}$.

Although it is not possible to determine decoupling with DTS temperature profiles alone, with the DTS temperature profiles we were able to study the canopy-atmosphere interaction in detail. However due to the fiber optic cables not being shielded or actively ventilated some uncertainty remains in the measured temperatures and gradients. This prevents conclusive interpretation in cases when temperature differences are small. A second shortcoming is the limited time resolution of the cables, which means that fast processes could not be studied. In future work we aim to use a thinner fiber for a fast thermal response could show more detail and unveil other processes which are not visible in the current data set. Finally, the current optical fiber technique may also be employed in a actively 'heated' form. In this 'hotwire modus' high resolution observation of wind speed is possible, as explained in Sayde et al. (2015), van Ramshorst et al. (2019), and Lapo et al. (2020). Such data would be complementary to the present study, and would give more insights into the wind shear and dynamic stability throughout the en-

tire canopy. This could aid in studying canopy-atmosphere interaction at forest sites, and would allow determining decoupling along the full height of the canopy. In turn this will increase the knowledge on the general drivers of decoupling in forests, to improve flux measurements above forests.

**Data availability.**

The data used in this study is available online on the 4TU data repository;

doi.org/10.4121/uuid:e0a3d8c9-cb3c-4029-bbe8-2b775c0b88ef, Schilperoort et al. (2020)

**Video supplement.**

Animation of all DTS temperature profiles used in this study are available online on Zenodo (Schilperoort et al., 2019).

**Author contributions.**

BS, CJR, and CT carried out the field measurements. BS processed the data and performed the data analysis. CJR provided site information. BS prepared the manuscript with contributions from all co-authors.

**Competing interests.**

The authors declare that they have no conflict of interest.

**Acknowledgements.**

The authors would like to thank Fred Bosveld (Royal Netherlands Meteorological Institute) for his comments on an earlier version of the manuscript, and for providing the psychrometer data at the Cabauw Experimental Site.

**Funding.**

This research was funded by NWO Earth and Life Sciences (ALW), veni-project 863.15.022, the Netherlands.

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

## Appendix A:  Speulderbos plant area index profile

To determine the vertical profile of the plant area index (PAI) at the research site, we took 10 images distributed over the height of the canopy, for 3 sides of the tower. The images were processed using Gap Light Analyser (Frazer et al., 1999). The results show that the bottom 20 m of the forest are bare, while the bulk of the branches and leaves are concentrated around 20 - 30 m
height (Fig. A1). The PAI as determined here is ~2.0, lower than the 4.5 LAI what Cisneros Vaca et al. (2018b) found in 2018. However the site experienced storms in 2018 and 2019, which removed a lot of branches and reduced the PAI.

## Appendix B:  Longwave radiation error estimation

The vertically placed DTS cables are not shielded or actively ventilated, and can therefore experience errors related to free radiative exposure. The error introduced by direct sunlight is quite significant and shielding is highly preferred to prevent
strong biases (Schilperoort et al., 2018). At night the cable can experience radiative cooling both to the sky and to the surface.

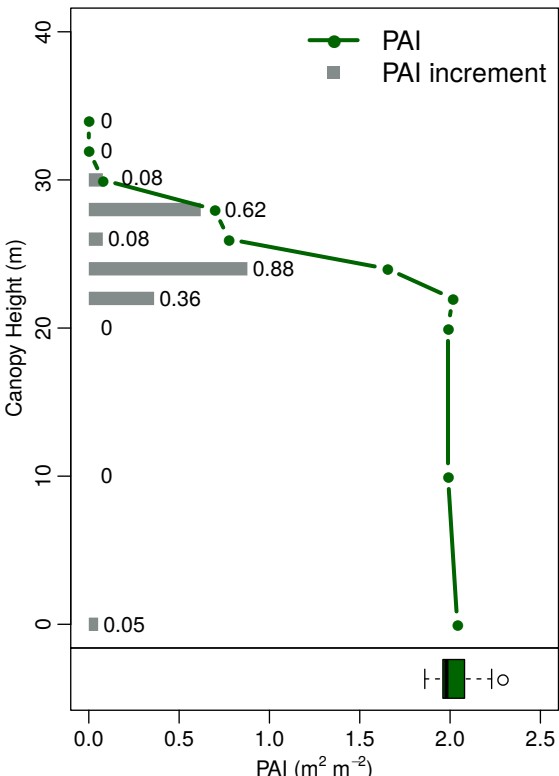

**Figure A1.** Profile of the plant area index (PAI) at the Speulderbos site.

If this longwave radiation error were be constant over the height, it would not affect the magnitude of the observed gradient. However, while the cooling rate is (close to) constant over height, the cable is warmed up by the surrounding air. This warming rate is dependent on the wind speed. As the wind speed varies over height, it will have a different warming rate at different heights, and thus create an error in the measured gradient.

## B1 Method

To estimate the radiation error on the DTS-measured vertical temperature gradients, data from a previous study at the Royal Netherlands Meteorological Institute's (KNMI) Cabauw Experimental Site for Atmospheric Research (CESAR) were used (Izett et al., 2019; Monna and Bosveld, 2013).

The measurements were set up in a grass field, maintained at  0.1 m height at the time of the experiment. Water-filled drainage ditches were at least 50 m from the setup. In the measurement field the FO cable was attached vertically to a hydraulic tower (Fig. B1). The DTS cable is of the same type as in the Speulderbos setup, and measured using the same Silixa Ultima-S

device. The DTS data was calibrated using a single-ended setup, with a fixed differential attenuation and temperature scaling parameter. A water bath located in a climate controlled room contained two loops of FO cable (outgoing and returning), and was used to determine the differential attenuation. The temperature scaling parameter was based on previous measurements
with the same cable.

At approximately 25 m from the hydraulic tower, a ventilated and shielded psychrometer setup was located, measuring the air temperature at 1, 2, and 4 m height. A WindMaster Pro 1352 sonic anemometer was located 6 m south from the hydraulic tower, with a measurement height of 0.6 m. The DTS temperatures at 1, 2, and 4 m height were calculated from the mean of the 3 data points near each height, e.g. the 3 data points within the range 0.8 to 1.2 m, to reduce the measurement uncertainty.

The measurement period ran from 3 November 2017 to 23 November 2017. The temperature gradients calculated based on 10 minute mean temperatures. For analysis all data during and 30 minutes after rainfall were discarded. Only nighttime data was used (when the incoming shortwave radiation was below 5 W/m$^2$), to isolate the effect of the longwave radiation. When humidity exceeded 98%, the data was also discarded, to account for condensation during fog events. Data was binned based on the 0.6 m wind speed.

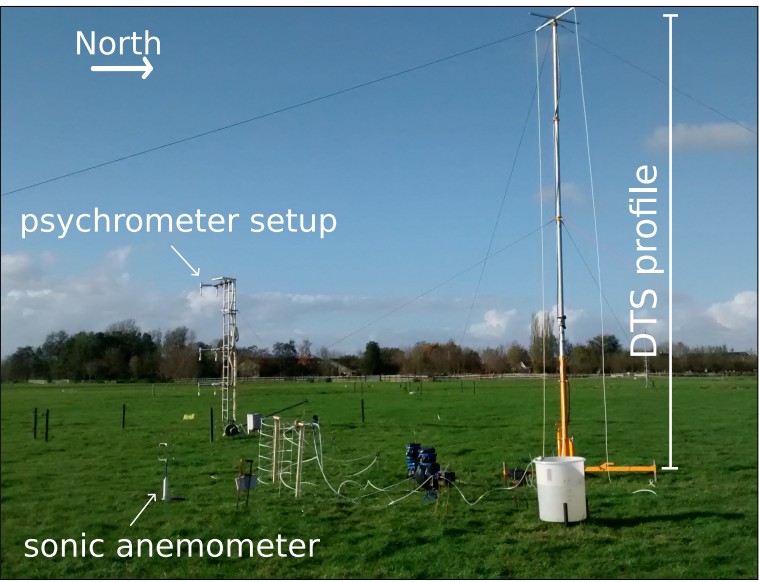

**Figure B1.** Overview of the measurement setup at CESAR

## B2  Absolute temperature error

To start we compare the absolute temperature of the FO cable to the psychrometer measured temperature (Fig. B2). The error is strongly dependent on the net longwave radiation and wind speed, with an error of up to 1.0 K during strong cooling and a lack of wind. Closer to the surface the error is larger (up to 1.5 K at 1 m), as the wind speed is lower there.

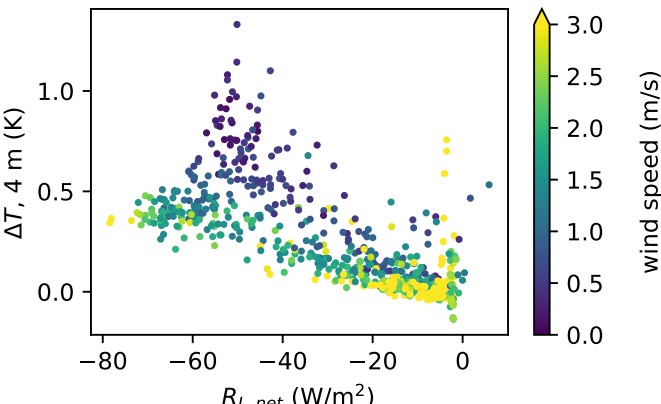

**Figure B2.** Difference between the psychrometer and DTS measured temperatures ($\Delta T$) at 4 m (positive numbers indicate a colder FO cable), compared to net longwave radiation and 0.6 m wind speed.

## B3  Gradient error

To calculate the gradients, the upper temperature is subtracted from the lower temperature. The DTS measurement error is calculated by subtracting the DTS-measured gradient from the psychrometer measured gradient.

For the gradient between 2 and 4 m, the error for higher wind speeds was below 0.01 K m$^{-1}$ across nearly the entire range of longwave radiation (Fig. B3). For low wind speeds, the error varied stronger with longwave radiation, and the error has a much higher variation. A reason for this high variation could be heterogeneity at the site during extremely stable atmospheric conditions.

The gradient between 1 and 4 m has a larger error compared to the 2 to 4 m gradient (Fig. B4), as the difference in wind speed between 1 and 4 m is much greater. For the lower range of cooling rate and with higher wind speeds, the error is not much greater than the lapse rate correction. However, with strong cooling rates ($R_{L,net} > 20$ W m$^{-2}$) and a low wind speed, the error becomes very large.

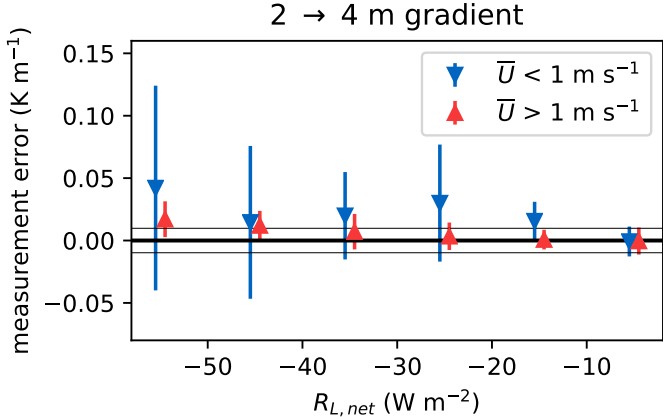

**Figure B3.** Error in the DTS-measured gradient between 2 and 4 m. Data are binned by the net longwave radiation. Split into 1 m wind speeds under 1 m/s (blue, n=207) and over 1 m/s (red, n=544). Grey lines indicate +/- 0.01 K. Error bars show +/- 1 standard deviation.

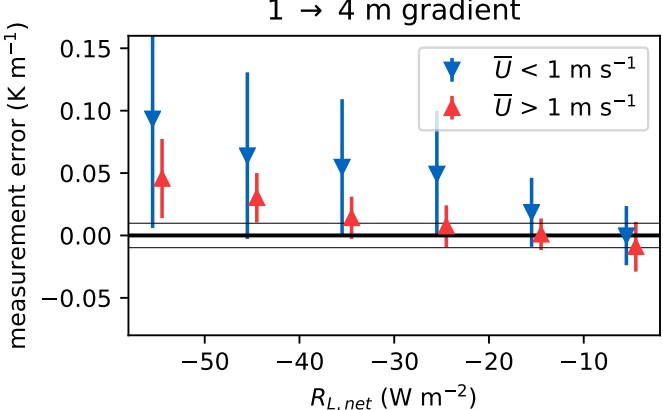

**Figure B4.** Error in the DTS-measured gradient between 1 and 4 m. Data are binned by the net longwave radiation. Split into 1 m wind speeds under 1 m/s (blue, n=207) and over 1 m/s (red, n=544). Grey lines indicate +/- 0.01 K. Error bars show +/- 1 standard deviation.