# Peer review of "Decoupling of a Douglas fir canopy: a look into the subcanopy with continuous vertical temperature profiles"

_Biogeosciences, 2020_

## Referee Comment (RC1) · Georg Jocher (Referee) · 7 Jul 2020

General evaluation: The manuscript deals with the exploration of canopy decoupling using the relatively new technique of temperature distributed sensing (DTS). As decoupling is a phenomenon relevant for each canopy and no standard method exists yet how to deal with it, the manuscript addresses a highly relevant scientific question using a novel approach. It fits well within the scope of the Journal Biogeosciences. The manuscript is well structured and written, easy understandable and conclusions are derived in a traceable manner. The presented results are sufficient to support the interpretations and conclusions. The title clearly reflects the content of the paper and

the abstract provides a concise and complete summary.

I have, however, three major suggestions regarding the current paper version:

It remains somewhat unclear to the reader, how the temperature error derivation procedure, obtained in a completely different environment than the measurement site, was transferred to the final setup. Please explain in more detail how you applied the error derivation procedure on the final data. Furthermore, it would be valuable to add few more words to the measurement principle of DTS itself to get information how the temperatures are obtained with this technique.

It would be great to set your whole work in a bit bigger context. There was already quite some work done on the topic decoupling, several different approaches developed. I miss the discussion of all the already existing work in the introduction. Once this is done, you can place your work in this context and explain how your work provides additional gain of knowledge in the context of the existing work.

I suggest to make use of all the data you have. If I understood this correctly, you have a sonic anemometer measuring within the canopy during the presented measurement period. Why not using these data too? With these data you can apply the approach by Thomas et al. (2013) who are assessing decoupling based on the relation of $\sigma$w above and below canopy, and compare these findings with your DTS data. This would give much additional value to your work. Furthermore, at one point you are mentioning advection and that you cannot assess it: I think you can. With the DTS data you can derive the buoyancy forcing which gives an indication regarding the potential of drainage flow near the surface. With the sonic anemometer you get wind direction and speed. Both quantities combined give you a clue, how important/relevant advection at your site could be (see Staebler and Fitzjarrald, 2004; 2005. Also Fig. 2 in Jocher et al., 2017).

Specific comments:

In the abstract (Line 10 etc.) it would be good to tell how you define decoupling. Which threshold of what you used for distinguishing between coupling and decoupling.

Lines 39 – 41: I don't think that you can say this generally, that decoupling occurs predominantly during daytime, while coupling during nighttime. It's rather the other way around. T profiles may indicate that, but the T profile is only one part of indicator for coupling or decoupling. The "nighttime" problem, i.e. underestimation of above canopy CO2 fluxes due to low turbulence and decoupling, is not called like that without reason (see e.g. Aubinet et al., 2012).

Lines 43 – 50: extend this part with the most important work and approaches on decoupling. Discuss also the implications of decoupling on above canopy derived fluxes bit more.

Lines 50 – 59: Great. This to compare with decoupling assessed by the correlation of $\sigma$w above and below canopy would be very valuable.

Lines 97 – 98: this refers to the understory measurements I assume?

Line 100 etc.: explain briefly the measurement principle of DTS.

Lines 122 – 127: how was this done in reality? You were grouping your data according specific conditions and applied then the error estimate on them which you derived from the reference setup? Explain this in detail.

Line 131: Best quality fluxes are fluxes with flag 0 only. Fluxes suitable for standard measurement programs are fluxes with flag 0 or 1. Specify.

Line 135: you introduce here the sonic. Why not using the data of this sensor?

Lines 205 etc.: how about counter-gradient fluxes? Fluxes against the gradient are possible, discuss this.

Lines 225 – 226: this is not possible to say in this way, that your u* threshold corresponds well with previous decoupling research, no proper justification. The u* threshold

is strongly site specific, and at certain sites it is even not possible to derive it.

Lines 245 – 246: somewhere else in the manuscript you are saying that radiation reaches the forest floor and heats it due to sparse vegetation, somehow this is contradictory.

Line 249: why restricting here the analysis on nighttime cases? In the introduction you are stating that decoupling occurs predominantly during daytime. I think it would be useful to make this analysis in 3.3.3 for both nighttime and daytime.

Line 277: you mention here that it would be interesting to explore the impact of understory stratification on the friction velocity threshold value by assessing effects of conditional sampling. Why not doing it here in this study?

Line 283: you are stating here that information of understory wind speed is lacking. But you have a sonic anemometer measuring in the canopy, so you would have this information ready. An analysis here combing the buoyancy forcing derived from DTS with wind speed and direction from the sonic anemometer can give you insights in potential drainage flow within the canopy.

References to consider:

Aubinet, M., Feigenwinter, C., Heinesch, B., Laffineur, Q., Papale, D., Reichstein, M., Rinne, J., van Gorsel, E., 2012. Nighttime flux correction. In: Aubinet, M., Vesala, T., Papale, D. (Eds.), Eddy Covariance: A Practical Guide to Measurement and Data Analysis. Springer, Dordrecht/Heidelberg/London/New York, pp. 133–157.

Staebler, R.M., Fitzjarrald, D.R., 2004. Observing subcanopy CO2 advection. Agric. For. Meteorol. 122, 139–156.

Staebler, R.M., Fitzjarrald, D.R., 2005. Measuring canopy structure and the kinematics of subcanopy flows in two forests. J. Appl. Meteorol. 44, 1161–1179.

---

## Referee Comment (RC2) · Anonymous Referee #2 · 15 Jul 2020

**General comments:**

This study uses temperature profiles with high vertical resolution measured within and above the canopy to investigate the issue of decoupling between the air within the canopy and the air above. This investigation is highly relevant, given that a decoupled situation will have significant impact on the exchanges of matter and energy between the canopy and the atmosphere, and it needs to be taken into account when interpreting field data. This type of temperature profile measurement is less common, and this study contributes to the assessment of its usefulness. The manuscript is clear, well-motivated and well-written, being suitable for publication at this journal. However, I

have three observations that I believe would help improve the impact of the manuscript.

First, I believe the study should be more precise in the definition and discussion of decoupling. Right now "decoupling" is defined in terms of the aerodynamic Richardson number, but throughout the text the term "decoupling" is used much more broadly, sometimes related to local static stability. This can be confusing and misleading in the conclusions. The second issue is related to the analysis of $u_*$ as an indicator of decoupling. In this specific analysis, I don't agree with the interpretation of the data and the conclusions (see details below). If a more precise definition of decoupling is used, maybe this analysis won't be needed. And finally, I believe that having a temperature profile instead of the typical point measurement of temperature should be taken more advantage of. Right now, the local stability and the convection height analyses are great examples of that, very interesting. But the temperature difference and aerodynamic Richardson number analyses use point temperature measurements, as in previous studies. I believe that here there is a big opportunity to improve these definitions, taking advantage of the profile measurements, as point measurements might not be the best reference of the entire layer temperature or even the decoupling. Everything that is happening between the two point measurements might be impacting the coupling state, and you should take advantage of having this information, convincing the reader that performing profile measurements are relevant (or not), compared to point measurements. Overall, I believe that the current static stability and convection height analyses, combined with a new temperature difference and decoupling definition that uses the temperature profiles, would provide a better analysis of decoupling and a more convincing discussion on the potential of temperature profile measurements.

**Specific comments:**

1. l. 50: "Instead of considering discrete point observations along the height of the canopy, we search for a more continuous probing of temperature to get a more

detailed view on decoupling along the entire height of the canopy." Can you elaborate more this paragraph? Can you discuss, for example, if previous studies on coupling have always used discrete observations, and if no study of coupling with continuous measurements have ever been done before? Maybe investigate the use of continuous measurements on decoupling of other atmospheric regions? I believe this should be the focus of the manuscript, and being more explicit would enhance the impression on the importance of the study.

2. Sec. 2.2: measurements other than temperature profile and $u_*$ were not used in this study (eddy covariance, ground and biomass heat flux, etc). Could they be used to infer coupling/decoupling? Maybe mention that they were present in the experiment, but not used here.

3. l. 164: "The height to which the parcel will rise is the height at which the local (potential) temperature $\theta(z)$ exceeds the temperature of the parcel." Which parcel? In the results it is mentioned the floor parcel, it should be clearer here.

4. Sec. 2.4: regarding the second order polynomial fit, why perform the fit of an analytical solution, but not use it to calculate the gradient analytically? If you are using finite-difference (eq. (3)), why not use it with the original data? How good are these fits? Can you show us some examples of the fit, to illustrate the level of quality? How about some statistics of the quality of the fit?

5. Sec. 2.4: you should emphasize that humidity effects are not taken into account (probably due to the lack of data) but that they could be relevant in this environment (if they are).

6. l. 175: "For the calculation of the aerodynamic Richardson number, we used the 10 m DTS temperature as the canopy internal temperature and the 44 m temperature as the top-of-canopy temperature." Why those specific heights? Shouldn't you take advantage of the fact that you have an entire profile?

7. Figure 4: can you add the polynomial fit in this figure as an example?

8. Sec. 3.3: "Dynamic and static decoupling" in the methods section you defined dynamic and static "stability", and mentioned "decoupling" only in the dynamic sense. Can you elaborate on the idea of "static decoupling"? Is it the same as "static stability"?

9. Sec. 3.3.1: I believe the comparison between $u_*$ and local temperature gradient is difficult due to the complex dynamics and the "cause" versus "consequence" misleading interpretation. I believe that as a first order approximation, we could think of $u_*$ and surface heat flux as causes, and temperature gradient as a consequence. But the temperature gradient will also impact $u_*$ and local heat flux. For that reason, comparing $u_*$ and temperature gradient directly can be misleading. For example, we can have high shear destroying temperature gradient (as discussed in this section), but we can also have low shear and low surface heat flux resulting in (and being a result of) low temperature gradient. It is a complex interplay and I don't think that looking for a threshold value is appropriate. The data in Fig. 7 a, b, c has more of a "L-shaped" curve than a proper "negative-correlation". It shows that at high shear it is impossible to sustain a large temperature gradient, but low-shear is actually concomitant with small and large temperature gradients. "At low shear conditions the top of the canopy is able to cool considerably, causing strong local gradients to occur." strong local gradients *can* occur, but will not necessarily occur. "Interesting, the understory gradients (Fig. 7b, c) show a characteristic behavior with a kind 'threshold' value for $u_*$: below $u_*$ large gradients tend to occur, while small gradients are observed for large $u_*$" I don't agree with this interpretation. Below $u_*$ in Fig. 7 b, c most of the data has small temperature gradients. "The forest floor is unstably stratified when $u_*$ is low, and stably stratified when $u_*$ is high", again, I believe there is too much dispersion in the data for this affirmation. "The strong relationships between the understory gradients and friction velocity show that the temperature gradients can serve as a proxy

for decoupling; when the friction velocity is low the understory is strongly stably stratified." as I said, I don't think there is a "strong relationship", and when the friction velocity is low the understory *can be* strongly stably stratified, but it won't be most of the time (I believe, based on the density of points in the figures). I suggest you improve this analysis and be more conservative in the discussion. If you want to keep this analysis (which I'm not sure it is needed), maybe you can use the thresholds of stability and the chosen thresholds of $u_*$ and count the number of occurrences in each category, providing a proportionality analysis such as the one in Fig. 5. Also add lines for those thresholds in Fig. 7 to help the visual interpretation of the data.

10. l. 230: "However, the understory can still be dynamically decoupled even without strong thermal stratification, as shown by the data points in the lower left corners of Fig.7b, c. It is likely that at very low friction velocities the wind will not be able to mix the canopy even though there is no strong temperature gradient (e.g., low wind, overcast conditions)." How do you know about the level of dynamical coupling from this analysis? You defined dynamical coupling from $Ri_A$, but it is not used here. How do you know that the data points in the lower left corners are dynamically decoupled? Can you be more precise in the definition of "dynamically (ou statically) decoupling", and include that in the figure? Maybe it will correspond to a region of the plot, maybe it will be a third variable, that can be added as colored dots in the plot.

11. l. 234: "While at night turbulent mixing is driven by wind shear (hence friction velocity), during daytime convection is also important for generating turbulence." Do you mean above the canopy?

12. Sec. 3.3.2 and 3.3.3: These analyses use temperature values defined at specific heights (44, 10 and 2m) to compare temperature differences within and above canopy, and to define an aerodynamic Richardson number and decoupling. This

was done as in a previous study at the same site (Bosveld et al. 1999), and although I think the direct comparison is useful and should be kept, I believe these analyses are not taking advantage of the temperature profile available. Could you replace these definitions by a more well-defined temperature difference (maybe some bulk or integrated temperature within each region) and to use a Richardson number that takes advantage of the temperature profile, or a decoupling definition that takes into account the information of the entire canopy? I believe that the definitions used by Bosveld et al. (1999) were chosen due to the data availability (point temperature), and here you have the opportunity to use a much more complete information with the temperature profile. Maybe there is a more suitable decoupling definition that takes into account the stability of the entire region (maybe in the literature about other parts of the atmosphere where temperature profiles are typically measured), something in the lines of the convection height analysis done here.

13. l. 253: "According to Bosveld et al. (1999), decoupling occurs when the aerodynamic Richardson number exceeds approximately 2." Since this decoupling criterion is used here, it is important to explain how it was obtained in the original study, and why it is also applicable here. It would be interesting to add that discussion to a definition of decoupling in the Methods section.

**Technical corrections:**

- Sec. 3.3.2: "Temperature difference subcanopy" improve title

- l. 306: "aN open subcanopy"

---

## Referee Comment (RC3) · Anonymous Referee #3 · 17 Jul 2020

General comments:

This manuscript deals with the decoupling in atmospheric boundary layer in a forest canopy, through the identification of static stability, from temperature profile obtained by the DTS technique. Forest canopy studies, which contain higher vertical resolution, are rare and may contribute to understanding the exchanges between under-canopy/canopy/free atmosphere above. Particularly, in very stability, conditions, the decoupling of layers under-canopy induce the accumulation, important in quantifying the exchange of momentum, water and scalars between the forest atmosphere, because contributes this balance. The manuscript add understanding of the flow over

forests and is well written, succinct and well organized. However, I have some considerations (suggestions): - I mainly suggest use the instruments at 0.8 ∼ 1m installed, in some way (sonic anemometer). You can use both for u* analysis, as well include others turbulent parameters, such $\sigma$w or VTKE, in relationships with temperature gradients via DST technique. (If the measurement period coincides). - A second methodology to determine decoupling thresholds of layers can be interesting, reinforcing your results. Either for all period, or maybe in case study (same periods used in section 3.1). I believe this is feasible, if high frequency measurements are available in anemometer (Gill 3D), in the specific comments I better present this suggestion.

Specific comments:

Lines 15 – 16: "This points towards the understory layer acting as a kind of mechanically 'blocking layer' between the forest floor and overstory", in fact I believe that a dense canopy, the leaves can act as turbulence filter. For that, it would be necessary adjust the time window of averages (in this case you used 15 min. I may be wrong!), to better observe this filtering. Some studies in forests have shown the turbulence in time scales until 100 seconds is restricted within canopy, while movements with larger scales can reach top and pass to above. Please consider adding something related to this.

Lines 39 – 40: "These regimes vary per site and are dependent on both the forest structure and the ambient weather conditions. In particular, the subcanopy tends to be decoupled during the day, when highest temperatures are found at the top of the canopy, and to be coupled in the night when lowest temperature occurs at the canopy top". The layer under canopy decoupled from the atmosphere above forest, generally, at night. You need review, because it's confused, or you be referring only the layers within canopy?

Section 2.4: Using polynomial fit, could you expose example of the profiles/gradients from raw data and after being adjusted. Maybe, can determine differents Richardson

numbers, taking advantage the temperature profile. One stability parameter above and another within the forest. Consider using the bulk Richardson number. (MAHRT, et al., 2013).

Lines 189 -190: "This will cause a stable stratification above the canopy and above the forest floor, while the bulk of the canopy (2 - 26 m) is unstably stratified due to the colder air in the overstory." I don't think 2-26m is unstable, but rather, near-neutral.However, if the classification was unstable, show the temperature gradient quantification that led this classification, it seems is very subtle.

Section 3.2: About forest floor discussion, is interesting analyzes between temperature gradient and friction speed at 1 m (sonic anemometer). Maybe, extrapolate using turbulence at level for other analyzes.

Section 3.3 - Also with the eddy covariance (48m) and sonic anemometer (0.8 ∼ 1m) systems, you can use some other turbulent parameters, perhaps $\sigma$w or VTKE (VTKE = 0.5 ($\sigma$uˆ2 + $\sigma$vˆ2 + $\sigma$wˆ2) ˆ1/2), in temperature gradients classification. If you choose VTKE, its relation with the average wind (could compare with the wind above and within canopy), can help determining threshold at under-canopy layer starts to be decoupled from levels above (see: SUN et al., 2012, ACEVEDO, et al. 2016).

Technical corrections:

line 95: "mean speed speed" double.

References:

ACEVEDO, O. C.; MAHRT, L.; PUHALES, F. S.; COSTA, F. D.; MEDEIROS, L. E.; DEGRAZIA, G. A. Contrasting structures between the decoupled and coupled states of the stable boundary layer. Quarterly Journal of the Royal Meteorological Society, Wiley Online Library, v. 142, n. 695, p. 693–702, 2016.

MAHRT, L.; THOMAS, C.; RICHARDSON, S.; SEAMAN, N.; STAUFFER, D.; ZEEMAN, M. Non-stationary generation of weak turbulence for very stable and weak-wind

conditions. Boundary-layer meteorology, Springer, v. 147, n. 2, p. 179–199, 2013.

SUN, J.; MAHRT, L.; BANTA, R. M.; PICHUGINA, Y. L. Turbulence regimes and turbulence intermittency in the stable boundary layer during cases-99. Journal of the Atmospheric Sciences, v. 69, n. 1, p. 338–351, 2012.

---

## Author Comment (AC1) · 26 Aug 2020

**The referee comments are copied in *blue*, our reply is in black**
* * *
Dear Georg Jocher,

Thank you for your comments on our manuscript. Based on your comments we hope to clarify what is unclear and improve the quality of the paper.

*General evaluation: The manuscript deals with the exploration of canopy decoupling using the relatively new technique of temperature distributed sensing (DTS). As decoupling is a phenomenon relevant for each canopy and no standard method exists yet how to deal with it, the manuscript addresses a highly relevant scientific question using a novel approach. It fits well within the scope of the Journal Biogeosciences. The manuscript is well structured and written, easy understandable and conclusions are derived in a traceable manner. The presented results are sufficient to support the interpretations and conclusions. The title clearly reflects the content of the paper and the abstract provides a concise and complete summary.*

*I have, however, three major suggestions regarding the current paper version:*

*It remains somewhat unclear to the reader, how the temperature error derivation procedure, obtained in a completely different environment than the measurement site, was transferred to the final setup. Please explain in more detail how you applied the error derivation procedure on the final data.*

Our intention was to get an order of magnitude estimation of the error caused by radiative cooling of the fiber optic cable. While the environments were indeed completely different, we transfer the results by looking at similar meteorological conditions (i.e., low wind speeds and a net longwave radiation $< 20$ W m$^{-1}$). Under those conditions the error in the gradient can be expected to be in the order of 0.01 K m$^{-1}$, an error we deemed to be acceptable. We did not correct or adjust the measured gradients, but left them as is. We will add a more detailed explanation when revising the manuscript.

*Furthermore, it would be valuable to add few more words to the measurement principle of DTS itself to get information how the temperatures are obtained with this technique.*

We will add an explanation of the measurement principle of DTS to the setup section.

*It would be great to set your whole work in a bit bigger context. There was already quite some work done on the topic decoupling, several different approaches developed. I miss the discussion of all the already existing work in the introduction. Once this is done, you can place your work in this context and explain how your work provides additional gain of knowledge in the context of the existing work.*

We will expand the introduction to properly put this work in the bigger context, discussing, e.g., *u\** filtering, *sigma_w* correlation, telegraphic approximation of *w*, and cross-correlation maximum between above and below canopy *w*.

*I suggest to make use of all the data you have. If I understood this correctly, you have a sonic anemometer measuring within the canopy during the presented measurement period. Why not using these data too? With these data you can apply the approach by Thomas et al. (2013) who are assessing decoupling based on the relation of σw above and below canopy, and compare these findings with your DTS data. This would give much additional value to your work.*

We did place a sonic anemometer at the bottom of the tower, but it only worked for a very short period of time before the equipment failed. I see now that this is currently not clearly explained in the manuscript. You are correct that if the data were available it could have added a lot of value to this work

*Furthermore, at one point you are mentioning advection and that you cannot assess it: I think you can. With the DTS data you can derive the buoyancy forcing which gives an indication regarding the potential of drainage flow near the surface. With the sonic anemometer you get wind direction and speed. Both quantities combined give you a clue, how important/relevant advection at your site could be (see Staebler and Fitzjarrald, 2004; 2005. Also Fig. 2 in Jocher et al., 2017)*

This would be a great addition and we will mention this possibility for future research, but sadly, without the data from the sonic anemometer (lacking as previously mentioned) we can not do this analysis.

*Specific comments:*

*In the abstract (Line 10 etc.) it would be good to tell how you define decoupling. Which threshold of what you used for distinguishing between coupling and decoupling.*

We will specify that we used the aerodynamic Richardson number to distinguish between coupling and decoupling.

*Lines 39 – 41: I don't think that you can say this generally, that decoupling occurs predominantly during daytime, while coupling during nighttime. It's rather the other way around. T profiles may indicate that, but the T profile is only one part of indicator for coupling or decoupling. The "nighttime" problem, i.e. underestimation of above canopy CO2 fluxes due to low turbulence and decoupling, is not called like that without reason (see e.g. Aubinet et al., 2012).*

We will change this sentence, and specifically mention the 'nighttime problem'.

*Lines 43 – 50: extend this part with the most important work and approaches on decoupling. Discuss also the implications of decoupling on above canopy derived fluxes bit more.*

In the revised manuscript will will extend this part of the introduction and place this work in a better context.

*Lines 50 – 59: Great. This to compare with decoupling assessed by the correlation of σw above and below canopy would be very valuable.*

Sadly the understory sonic anemometer failed, and the collected data is insufficient and does not overlap with the DTS measurements.

*Lines 97 – 98: this refers to the understory measurements I assume?*

Indeed it does. We will change the sentence to make this more clear and less ambiguous.

*Line 100 etc.: explain briefly the measurement principle of DTS.*

We will add an explanation of the measurement principle of DTS to the setup section.

*Lines 122 – 127: how was this done in reality? You were grouping your data according specific conditions and applied then the error estimate on them which you derived from the reference setup? Explain this in detail.*

We did not correct for any radiation errors, but only used the results from the Cabauw measurements to make an estimation of the possible error or bias in the results. As the error under the expected conditions was only in the order of $0.01 \text{ K m}^{-1}$ we regarded this error as acceptable.

*Line 131: Best quality fluxes are fluxes with flag 0 only. Fluxes suitable for standard measurement programs are fluxes with flag 0 or 1. Specify.*

We will change this sentence to "These flags represent fluxes suitable for general analysis, ...", removing "are the best quality fluxes"

*Line 135: you introduce here the sonic. Why not using the data of this sensor?*

As mentioned before, we did place a sonic anemometer at the bottom of the tower, but it only worked for a very short period of time before the equipment failed. We will state this clearly in the revised manuscript.

*Lines 205 etc.: how about counter-gradient fluxes? Fluxes against the gradient are possible, discuss this.*

Counter-gradient fluxes are indeed possible when, for example, the air temperature above the forest floor is higher than the temperature above the

forest, but lower than the temperature within the overstory. We will add a sentence here to discuss counter-gradient fluxes.

*Lines 225 – 226: this is not possible to say in this way, that your u\* threshold corresponds well with previous decoupling research, no proper justification. The u\* threshold is strongly site specific, and at certain sites it is even not possible to derive it.*

Will will change this sentence and mention the issues with u\* thresholds; "...corresponds to results from ..., although the u\* threshold is strongly site specific, and it is not always possible to derive a u\* threshold."

*Lines 245 – 246: somewhere else in the manuscript you are saying that radiation reaches the forest floor and heats it due to sparse vegetation, somehow this is contradictory.*

Very little light penetrates the canopy to reach the forest floor. Some can filter through to warm up the forest floor slightly, but this is only a small fraction of the total incoming sunlight.

*Line 249: why restricting here the analysis on nighttime cases? In the introduction you are stating that decoupling occurs predominantly during daytime. I think it would be useful to make this analysis in 3.3.3 for both nighttime and daytime.*

For section 3.3.3 we restricted the analysis to nighttime cases as the underlying assumptions of the aerodynamic Richardson number we used are not valid for daytime conditions. The friction velocity will be strongly affected by turbulence generated by convection from the top of the canopy, and is therefore not a good measure of the wind shear mixing the under-story from the top down. The suppression of mixing by the stable stratification of the understory is also not included into the aerodynamic Richardson number.

*Line 277: you mention here that it would be interesting to explore the impact of understory stratification on the friction velocity threshold value by assessing effects of conditional sampling. Why not doing it here in this study?*

Due the inconclusive results and lack of data we chose to not include this analysis in the manuscript. As a demonstration the plot below shows the mean vector in every bin; i.e. where the next data point (in time) would be. The bins are denoted by the gray boxes.

[Figure]

With the data available some slight patterns can be seen, but the uncertainty large and there is still a lack of sufficient data to get a clear pattern. If more data were available more filtering could be performed, e.g., for clear sky conditions, and a more conclusive picture could form.

*Line 283: you are stating here that information of understory wind speed is lacking. But you have a sonic anemometer measuring in the canopy, so you would have this information ready. An analysis here combing the buoyancy forcing derived from DTS with wind speed and direction from the sonic anemometer can give you insights in potential drainage flow within the canopy.*

As mentioned previously, the sonic anemometer only worked for a short period of time before failing. Without this data I think we can not easily get a better insight in the drainage flow within the canopy.

---

## Author Comment (AC2) · 26 Aug 2020

**The referee comments are copied in *blue*, our reply is in black**
* * *
*This study uses temperature profiles with high vertical resolution measured within and above the canopy to investigate the issue of decoupling between the air within the canopy and the air above. This investigation is highly relevant, given that a decoupled situation will have significant impact on the exchanges of matter and energy between the canopy and the atmosphere, and it needs to be taken into account when interpreting field data. This type of temperature profile measurement is less common, and this study contributes to the assessment of its usefulness. The manuscript is clear, well-motivated and well-written, being suitable for publication at this journal. However, I have three observations that I believe would help improve the impact of the manuscript.*

*First, I believe the study should be more precise in the definition and discussion of decoupling. Right now "decoupling" is defined in terms of the aerodynamic Richardson number, but throughout the text the term "decoupling" is used much more broadly, sometimes related to local static stability. This can be confusing and misleading in the conclusions.*

*The second issue is related to the analysis of u\* as an indicator of decoupling. In this specific analysis, I don't agree with the interpretation of the data and the conclusions (see details below). If a more precise definition of decoupling is used, maybe this analysis won't be needed.*

*And finally, I believe that having a temperature profile instead of the typical point measurement of temperature should betaken more advantage of. Right now, the local stability and the convection height analyses are great examples of that, very interesting. But the temperature difference and aerodynamic Richardson number analyses use point temperature measurements, as in previous studies. I believe that here there is a big opportunity to improve these definitions, taking advantage of the profile measurements, as point measurements might not be the best reference of the entire layer temperature or even the decoupling. Everything that is happening between the two point measurements might be impacting the coupling state, and you should take advantage of having this information, convincing the reader that performing profile measurements are relevant (or not), compared to point measurements. Overall, I believe that the current static stability and convection height analyses, combined with a new temperature difference and decoupling definition that uses the temperature profiles, would provide a better analysis of decoupling and a more convincing discussion on the potential of temperature profile measurements.*

Dear referee,

Thank you for taking the time to look at and comment on our manuscript. We will try to be more clear and consistent with our use of decoupling, and not use it in relation to just the static stability. As point two and three return in the specific comments, we will reply to them there.

*Specific comments:*
*1. l. 50: "Instead of considering discrete point observations along the height of the canopy, we search for a more continuous probing of temperature to get a more detailed view on decoupling along the entire height of the canopy." Can you elaborate more this paragraph? Can you discuss, for example, if previous studies on coupling have always used discrete observations, and if no study of coupling with continuous measurements have ever been done before? Maybe investigate the use of continuous measurements on decoupling of other atmospheric regions? I believe this should be the focus of the manuscript, and being more explicit would enhance the impression on the importance of the study.*

Continuous measurements in the atmosphere have been performed with different equipment on a much larger scale (e.g. radar or sodar wind profiling). Within canopies we are not aware of any previous studies with continuous measurements, although some field campaigns have had a very high density of instrumentation, e.g., the Canopy Horizontal Turbulence Array (CHATS).
In the revised manuscript we will expand the paragraph and discuss previous studies in more detail.

*2. Sec. 2.2: measurements other than temperature profile and u\* were not used in this study (eddy covariance, ground and biomass heat flux, etc). Could they be used to infer coupling/decoupling? Maybe mention that they were present in the experiment, but not used here*

The other measurements are indeed not used in this study, except for the general characterization of the energy balance in the understory. I do not think the ground and biomass heat fluxes can (easily) be used to infer decoupling. If all energy balance components could be measured without uncertainty, coupling/decoupling could be inferred from that, but the uncertainty in these measurements is too high.
We will make it more clear in the revised manuscript that the measurements were present but not used extensively.

*3. l. 164: "The height to which the parcel will rise is the height at which the local (potential) temperature $\theta(z)$ exceeds the temperature of the parcel." Which parcel? In the results it is mentioned the floor parcel, it should be clearer here.*

We will clarify this by specifically calling it the 'floor parcel';
"The height to which, e.g., a parcel of air from the forest floor will rise is the height at which the local (potential) temperature $\theta(z)$ exceeds the temperature of the forest floor parcel."

*4. Sec. 2.4: regarding the second order polynomial fit, why perform the fit of an analytical solution, but not use it to calculate the gradient analytically? If you are using finite-difference (eq. (3)), why not use it with the original data? How good are these fits? Can you show us some*

*examples of the fit, to illustrate the level of quality? How about some statistics of the quality of the fit?*

We chose to calculate a polynomial fit through the data points as DTS suffers from (normally distributed) measurement noise. By calculating the fit a lot of this noise will be filtered out, and the resulting temperature gradient can be calculated more accurately.
In the figure below the polynomial fits are added to the data of Figure 4a.

[Figure]

As an indication of the goodness of the fits, we calculated the RMSE of the fit, the mean values are shown in the table below. Note that the RMSE of the fit is also influenced by the measurement uncertainty of DTS.

| Profile | Mean RMSE of polynomial fit (K) |
| --- | --- |
| Above-canopy | 0.095 |
| Overstory | 0.087 |
| Upper-understory | 0.120 |
| Lower-understory | 0.112 |
| Forest floor | 0.022 |

In the revised manuscript we will shortly discuss the goodness of the fits in the Data Processing section.

*5. Sec. 2.4: you should emphasize that humidity effects are not taken into account(probably due to the lack of data) but that they could be relevant in this environment (if they are).*

Humidity effects are indeed relevant, and could assist in transport from the forest floor or understory to the atmosphere above. In this study we looked at the potential temperature only taking into account the dry adiabatic lapse rate. We will emphasize this in the revised manuscript.

*6. l. 175: "For the calculation of the aerodynamic Richardson number, we used the 10 m DTS temperature as the canopy internal temperature and the 44 m temperature as the top-of-canopy temperature." Why those specific heights? Shouldn't you take advantage of the fact that you have an entire profile?*

The 10 m temperature is in the center of the understory, and therefore represents the general temperature of the understory well. It could be possible to use a sample of different heights, or a profile integrated temperature but this will not change the results significantly.
The 44 m temperature was chosen as this is close to the sonic anemometer which provides the data for u*.

*7. Figure 4: can you add the polynomial fit in this figure as an example?*

Figure 4 with the polynomial fits added is shown in the answer to question 4. We prefer not to add the fits to the manuscript as it would distract from the goal of Figure 4.

*8. Sec. 3.3: "Dynamic and static decoupling" in the methods section you defined dynamic and static "stability", and mentioned "decoupling" only in the dynamic sense. Can you elaborate on the idea of "static decoupling"? Is it the same as "static stability"?*

This is indeed not clear. We will change the section title to "Influence of dynamic and static stability on decoupling"

*9. Sec. 3.3.1: I believe the comparison between u\* and local temperature gradient is difficult due to the complex dynamics and the "cause" versus "consequence" misleading interpretation. I believe that as a first order approximation, we could think of u\* and surface heat flux as causes, and temperature gradient as a consequence. But the temperature gradient will also impact u\* and local heat flux. For that reason, comparing u\* and temperature gradient directly can be misleading. For example, we can have high shear destroying temperature gradient (as discussed in this section), but we can also have low shear and low surface heat flux resulting in (and being a result of) low temperature gradient. It is a complex interplay and I don't think that looking for a threshold value is appropriate.*

*The data in Fig. 7 a, b, c has more of a "L-shaped" curve than a proper "negative-correlation". It shows that at high shear it is impossible to sustain a large temperature gradient, but low-shear is actually concomitant with small and large temperature gradients."At low shear conditions the top of the canopy is able to cool considerably, causing strong local gradients to occur." strong local gradients can occur, but will not necessarily occur.*

*"Interesting, the understory gradients (Fig. 7b, c) show a characteristic behavior with a kind 'threshold' value for u\*: below u\*=0.4 large gradients tend to occur, while small gradients are observed for large u\*" I don't agree with this interpretation. Below u\*=0.4 in Fig. 7 b, c most of the data has small temperature gradients.*

*"The forest floor is unstably stratified when u\* is low, and stably stratified when u\* is high", again, I believe there is too much dispersion in the data for this affirmation. "The strong relationships between the understory gradients and friction velocity show that the temperature gradients can serve as a proxy for decoupling; when the friction velocity is low the understory is strongly stably stratified." as I said, I don't think there is a "strong relationship", and when the friction velocity is low the understory can be strongly stably stratified, but it won't be most of the time (I believe, based on the density of points in the figures).*

*I suggest you improve this analysis and be more conservative in the discussion. If you want to keep this analysis (which I'm not sure it is needed), maybe you can use the thresholds of stability and the chosen thresholds of u\* and count the number of occurrences in each category, providing a proportionality analysis such as the one in Fig. 5. Also add lines for those thresholds in Fig. 7 to help the visual interpretation of the data.*

Indeed strong gradients will not necessarily occur. Besides u*<0.4, the conditions have to be right to allow for cooling of the canopy (i.e., clear skies). We will make it more clear in the revised manuscript that u* alone is not enough to discern decoupling, and refrain from calling it a hard threshold.

In our answer to your comment #10 we have added a plot which shows the two regimes (coupled and decoupling), and them overlapping in the lower left corner of the plot. This shows that even if u*<0.4 the canopy can indeed still be coupled.

We will revise this section and will more more conservative and clear in our interpretation.

> *10. l. 230: "However, the understory can still be dynamically decoupled even without strong thermal stratification, as shown by the data points in the lower left corners of Fig.7b, c. It is likely that at very low friction velocities the wind will not be able to mix the canopy even though there is no strong temperature gradient (e.g., low wind, overcast conditions)." How do you know about the level of dynamical coupling from this analysis? You defined dynamical coupling from RiA, but it is not used here. How do you know that the data points in the lower left corners are dynamically decoupled? Can you be more precise in the definition of "dynamically (or statically) decoupling", and include that in the figure? Maybe it will correspond to a region of the plot, maybe it will be a third variable, that can be added as colored dots in the plot.*

Thank you for the suggestion to add RiA to the plots as colored dots. In the image below you can see the result for the nighttime temperature gradient in the upper-understory (i.e., the data of Figure 7b).

[Figure]

In the figure the color jumps at RiA=2; which was the decoupling threshold found by Bosveld et al. (1999). The two regimes (either coupled or decoupled) are very clearly visible. The regimes overlap around dθ/dz=0 and u*<0.4. We will add the RiA color coded dots to Figure 7a,b,c,d, and discuss this in the revised manuscript.

*11. l. 234: "While at night turbulent mixing is driven by wind shear (hence friction velocity), during daytime convection is also important for generating turbulence." Do you mean above the canopy?*

Yes, we mean convection at the top of the canopy and above. We will make this more clear.

*12. Sec. 3.3.2 and 3.3.3: These analyses use temperature values defined at specific heights (44, 10 and 2m) to compare temperature differences within and above canopy, and to define an aerodynamic Richardson number and decoupling. This was done as in a previous study at the same site (Bosveld et al. 1999), and although I think the direct comparison is useful and should be kept, I believe these analyses are not taking advantage of the temperature profile available. Could you replace these definitions by a more well-defined temperature difference (maybe some bulk or integrated temperature within each region) and to use a Richardson number that takes advantage of the temperature profile, or a decoupling definition that takes into account the information of the entire canopy? I believe that the definitions used by Bosveld et al. (1999) were chosen due to the data availability (point temperature), and here you have the opportunity to use a much more complete information with the temperature profile. Maybe there is a more suitable decoupling definition that takes into account the stability of the entire region(maybe in the literature about other parts of the atmosphere where temperature profiles are typically measured), something in the lines of the convection height analysis done here.*

We can change the temperatures involved to the bulk/integrated temperatures for each region (e.g. forest floor and lower understory), however this will not impact the results in any significant way. To illustrate this the image below shows Figure 9, with added data of the 2 m and 15 m temperatures instead of only the 10 m temperature. While there are differences, these are not very large.

[Figure]

A decoupling definition that takes into account the entire temperature profile does sound very interesting, but we are not entirely sure on how to approach this.

*13. l. 253: "According to Bosveld et al. (1999), decoupling occurs when the aerodynamic Richardson number exceeds approximately 2." Since this decoupling criterion is used here, it is important to explain how it was obtained in the original study, and why it is also applicable here. It would be interesting to add that discussion to a definition of decoupling in the Methods section.*

We will add an explanation to the Methods section on how Bosveld et al. determined the critical aerodynamic Richardson number.

*Technical corrections:*
*Sec. 3.3.2: "Temperature difference subcanopy" improve title*

We will change the title to "Magnitude of the temperature difference between the subcanopy and atmosphere"

*l. 306: "aN open subcanopy"*

Thank you. This has been corrected.

---

## Author Comment (AC3) · 26 Aug 2020

**The referee comments are copied in *blue*, our reply is in black**
* * *
*General comments:*
*This manuscript deals with the decoupling in atmospheric boundary layer in a forest canopy, through the identification of static stability, from temperature profile obtained by the DTS technique. Forest canopy studies, which contain higher vertical resolution, are rare and may contribute to understanding the exchanges between under-canopy/canopy/free atmosphere above. Particularly, in very stability, conditions, the decoupling of layers under-canopy induce the accumulation, important in quantifying the exchange of momentum, water and scalars between the forest atmosphere, because contributes this balance. The manuscript add understanding of the flow over forests and is well written, succinct and well organized. However, I have some considerations (suggestions): - I mainly suggest use the instruments at 0.8~1m installed, in some way (sonic anemometer). You can use both for u\* analysis, as well include others turbulent parameters, such σw or VTKE, in relationships with temperature gradients via DST technique. (If the measurement period coincides). - A second methodology to determine decoupling thresholds of layers can be interesting, reinforcing your results. Either for all period, or maybe in case study (same periods used in section 3.1). I believe this is feasible, if high frequency measurements are available in anemometer (Gill3D), in the specific comments I better present this suggestion.*

Dear referee,

Thank you for taking the effort to read our manuscript, and thank you for the compliments. Sadly the sonic anemometer at 1 m broke down and only functioned for a short period of time. This is not clearly stated in the manuscript and will be corrected.

*Specific comments:*

*Lines 15 – 16:  "This points towards the understory layer acting as a kind of mechanically 'blocking layer' between the forest floor and overstory", in fact I believe that a dense canopy, the leaves can act as turbulence filter. For that, it would be necessary adjust the time window of averages (in this case you used 15 min. I may be wrong!),to better observe this filtering. Some studies in forests have shown the turbulence in time scales until 100 seconds is restricted within canopy, while movements with larger scales can reach top and pass to above. Please consider adding something related to this.*

In this study we used time averages of 15 minutes due to the limitations of the measurement technique. With DTS it is currently not possible to measure gradients at the required precision on much smaller time scales. The response speed of the ables used in this study (up to 5 minutes in slow moving air) is also a limiting factor.

We will add information on time scale dependent turbulence filtering by the canopy to the discussion.

> *Lines 39 – 40: "These regimes vary per site and are dependent on both the forest structure and the ambient weather conditions. In particular, the subcanopy tends to be decoupled during the day, when highest temperatures are found at the top of the canopy, and to be coupled in the night when lowest temperature occurs at the canopy top". The layer under canopy decoupled from the atmosphere above forest, generally, at night. You need review, because it's confused, or you be referring only the layers within canopy?*

We will change this sentence to "… most commonly the subcanopy tends to be coupled during the day, and to be coupled in the night when the canopy cools down due to radiative cooling." and refer specifically to the 'nighttime problem' in the revised manuscript.

> *Section 2.4: Using polynomial fit, could you expose example of the profiles/ gradients from raw data and after being adjusted.*

In the image below the polynomial fits are plotted over the raw data of Figure 4a.

[Figure]

*Maybe, can determine differents Richardson numbers, taking advantage the temperature profile. One stability parameter above and another within the forest. Consider using the bulk Richardson number. (MAHRT, et al.,2013).*

We are sadly not able to calculate the bulk Richardson number as we do not have the required data (i.e., difference in horizontal wind components) due to the lack of continuous understory wind speed measurements.

*Lines 189 -190: "This will cause a stable stratification above the canopy and above the forest floor, while the bulk of the canopy (2 – 26 m) is unstably stratified due to the colder air in the overstory." I don't think 2 – 26 m is unstable, but rather, near-neutral. However, if the classification was*

*unstable, show the temperature gradient quantification that led this classification, it seems is very subtle.*

For the nighttime profile during the clear diurnal cycle we classified the 2 – 26 m section as stable due to the presence of the cold air in the overstory. The (potential) temperature difference between the overstory and understory is approximately 0.7 K, we think this is sufficient to classify that section as unstable.

*Section 3.2: About forest floor discussion, is interesting analyzes between temperature gradient and friction speed at 1 m (sonic anemometer). Maybe, extrapolate using turbulence at level for other analyzes.*
*Section 3.3 – Also with the eddy covariance (48 m) and sonic anemometer (0.8~1m) systems, you can use some other turbulent parameters, perhaps σw or VTKE (VTKE= 0.5 (σuˆ2 +σvˆ2 +σwˆ2) ˆ1/2), in temperature gradients classification. If you choose VTKE, its relation with the average wind (could compare with the wind above and within canopy), can help determining threshold at under-canopy layer starts to be decoupled from levels above (see: SUN et al., 2012, ACEVEDO, et al. 2016).*

We would have liked to have made these comparisons and calculations, but sadly the sonic anemometer at 1 m broke down and only functioned for a short period of time, not overlapping with the other sensors. This is not clearly stated in the manuscript and will be clarified in the revised version.

*Technical corrections:*
*line 95: "mean speed speed" double.*

Thank you. This has been corrected.

---

## Author Response (AR2)

Dear Paul Stoy,

Thank you for taking the time to be the editor of the review process.
Below are our point-by-point replies to the comments of the referees. We hope to have hereby addressed all their concerns. After the replies there is a short list of changes. Last is a marked up version of the manuscript, where all changes are shown.

Best regards,
Bart Schilperoort

**Reply to referees**

Referee comments are shown in italic, our replies in regular font.

**Referee #2**

*Technical corrections:*
*- l.54: "based on based on"*
This has now been corrected

*- l.173: "the temperature probes"*
Added capitalization to "The".

*- Eq.(3): why approximate the derivative using finite difference if you have a polynomial fit that gives the derivative directly?*
It is also possible to get the derivative directly from the polynomial fit, but calculating the finite difference was more straightforward in the code package used.

*- l. 256: "the results suggests"*
Corrected to "suggest".

*- l. 296: "Interestingly" instead of "Interesting"?*
This has now been corrected

*- l. 304: "correlation: The forest floor"*
Corrected to "*correlation: the forest floor*"

**Referee #1: Georg Jocher**

*Line 13: insert "to" so that is written there "…..and the canopy was able to cool down through…"*
This has now been corrected

*Line 47: insert "above-canopy" so that is written there "…the interpretation of on-site above-canopy flux measurements…"*
We have added "above-canopy" to the sentence.

*Line 54: replace "determined" with "approached"; replace "has been so-called" with "is the so-called".*
We have incorporated these suggestions.

*Line 54 and following: general comment to the u\* filtering: it was not developed to approach decoupling, that was not the initial aim of this procedure. The initial aim was to ensure sufficient turbulence for EC measurements. Decoupling issues are then addressed by this automatically in some way, but not completely. Revise this section accordingly.*
We have revised the start of this paragraph as such:
"In previous studies decoupling has been approached in a number of ways. A commonly used method for flux measurement quality control is the so-called `u\* filtering' . In this method data with low friction velocities (u\*) is flagged to ensure that there is sufficient turbulence for eddy covariance. This means decoupling issues are partially addressed automatically."

*Line 57: "…albeit with a higher value than a dynamic threshold would have." What does this mean? Not fully clear, clarify.*

We have changed this sentence to "These site specific stable u* values are higher than in studies where the threshold varied in time."

*Line 59: delete one of the two "based on".*

This has now been corrected

*Line 71: I would remove the Foken (2017) reference here. They used cross correlation in a different context, citing Jocher et al. (2020) here is sufficient.*

This has now been corrected

*Line 72: write "…an improved estimation of the fluxes above the canopy…*

We have incorporated this suggestion.

*Line 73: move the sentence "Usually, however, eddy covariance measurements are only available above the canopy." to the end of the paragraph.*

This sentences has been moved.

*Line 89: I wonder if it would be possible to name at the end of the introduction a couple of specific objectives of this study which are addressed then later in the manuscript. And close the introduction with the overall aim of the study.*

To address this comment we have changed the last paragraph to the following:
"The main goals of this study are to see how common decoupling is at the Speulderbos site, and to study the influence of shear and buoyancy on decoupling, especially along the height of the canopy.
By using high resolution DTS measured temperature profiles we aim to get a more detailed view on decoupling along the entire height of the canopy, the response of the atmosphere-canopy system, and if vertical mixing by turbulence is suppressed or enhanced due to thermal stratification."

*Line 119: delete the second "and". Two times section in this line. Other expression for the second one? Like part?*

This has been changed to "Subcanopy: the section between the ground and overstory, consisting of three parts"

*Line 125-130: again I am wondering why you are naming this all here if it is not used in the further course of the study. I would only introduce what is actually used in your work.*

We have removed "The ground heat… …heat flux"

*Line 160 (and other places in the manuscript): write in past tense when you describe what you did in the past. Tenses are somehow mixed here…*

This has been corrected.

*Lines 158-163: so that means in reality you were filtering the data according their potential error, discarding all data above a certain threshold? Name this explicitly.*

Under the conditions in the understory (low wind speeds, net longwave radiation $< 20 \text{ W/m}^2$), we expect the error in the measured gradients to be 0.01 K/m or smaller. As such we did not filter the data. We have adjusted the paragraph to to make this more clear.

*Line 170: No, only Flag 0 values are best quality data. Flags 0 and 1 are good enough for general analysis, but not best quality. Revise this.*

This sentence has been revised to "These flags represent fluxes suitable for general analysis, …"

*Lines 174-183: delete what you not specifically use in the study.*
We have removed the sensors we did not use elsewhere in the study.

*Line 204: reformulate in "…..diverged and consequently decoupling was assumed."*
We have incorporated this suggestion.

*Line 223: "…..was ~0.1 K for the main profile…"*
This has been corrected.

*Line 242: delete "due to radiative cooling." at the end of the sentence.*
This has been corrected.

*Line 250: how are seasons defined?*
We followed the meteorological temperate season definition. We have clarified this by adding the following sentence; "where winter is from December to February, spring from March to May, summer from June to August, and fall from September to November."

*Line 272: delete the last sentence.*
We have incorporated this suggestion.

*Line 303: "Interestingly,…"; "…with a kind of threshold value…"*
This has been corrected.

*Line 305: reformulate: "This threshold value is in line with…"*
We have incorporated this suggestion.

*Lines 354-364: this section says actually, that it is only the intensity of wind shear which dominates the coupling/decoupling behaviour. Is it this what you want to say? So why should we care about T profiles?*
The coupling/decoupling behavior is indeed mostly dominated by the intensity of wind shear, except for the lowest part of the canopy; just above the forest floor. To address this we have added the following sentence to the end of this paragraph:
"To determine if the subcanopy is fully coupled requires local measurements, either of a temperature profile or of turbulence."

*Line 365: "which is in line with previous research"*
We have incorporated this suggestion.

*Line 373: to the drainage flows; this was not found by Alekseychik et al., this knowledge exists already longer. Choose another reference maybe. Or reformulate:" Amongst others, Alekseychik et al. Proved….."*
We have reformulated this sentence to "Amongst others, Alekseychik et al. proved that drainage flows, …"

*Line 378: add reference Jocher et al. (2017) to the Staebler and Fitzjarrald reference.*
We have incorporated this suggestion.

*Line 400: "…response which could show…"*
This has been corrected.

*Line 406: revise text in "…to improve the representativeness of flux measurements above the forest."*

We have incorporated this suggestion.

> *Line 407: there was a lot of talking about decoupling now, very interesting. But unfortunately you almost don`t touch at all the implications of decoupling. Why would we be interested to know about decoupling? What can happen with the above canopy EC fluxes if decoupling occurs? 1-2 sentences to this would be great.*

To address this comment we have added the following paragraph to the start of the conclusion:
"Due to the tall vertical structure of forests, parts of the canopy can be turbulently decoupled from the overlying atmosphere.
If this decoupling is not properly taken into account in above-canopy flux measurements, it can lead to incorrect estimates of gas fluxes, in turn reducing the accuracy of estimates of the net ecosystem exchange of $CO_2$, or evaporation."

> *Line 576: "…would be constant over the height…"*

This has been corrected.

> *Line 595: "…gradients were calculated…"*

This has been corrected.

> *Line 605: why are you substracting here the upper from the lower temperature? For the T profiles you do it the other way around, right?*

This was incorrectly written down. We indeed did it the other way around (subtracting lower temperature from upper temperature). This is now corrected.

> *Line 608: replace "has" with "had".*

This has been corrected.

> *Lines 609/610: if there is heterogeneity, then it is always there, not only during stable conditions. revise this sentence.*

The heterogeneity is indeed always there. However, it will become more apparent during strongly stable conditions due to the reduced turbulent mixing. We have revised this sentence to "A reason for this high variation could be heterogeneity at the site, which has a stronger influence during extremely stable atmospheric conditions."

**List of relevant changes**

- Corrected spelling and grammar errors throughout the document

- Small changes in sentence structure and wording through the document

- A short statement on the goals of the study has been added to the end of the introduction

- Information from unused senors has been scrapped from the end of section 2.1

- Descriptions of unused sensors has been removed from the end of section 2.2

- At the start of the conclusions a paragraph has been added, explaining why knowing if decoupling occurs is relevant

[revised manuscript text omitted]